



# Natural emissions of VOC and NO$_x$ over Africa constrained by TROPOMI HCHO and NO$_2$ data using the MAGRITTEv1.1 model

Beata Opacka[1], Trissevgeni Stavrakou[1], Jean-François Müller[1], Isabelle De Smedt[1], Jos van Geffen[2], Eloise A. Marais[3], Rebekah P. Horner[3], Dylan B. Millet[4], Kelly C. Wells[4], Alex B. Guenther[5]

[1]Royal Belgian Institute for Space Aeronomy (BIRA-IASB), Brussels, 1180, Belgium
[2]Royal Netherlands Meteorological Institute (KNMI), De Bilt, the Netherlands
[3]Department of Geography, University College London, London, UK
[4]Department of Soil, Water, and Climate, University of Minnesota, St Paul, MN, USA
[5]Department of Earth System Science, University of California Irvine, 92697, California, USA

*Correspondence to*: Beata Opacka (beata.opacka@aeronomie.be) and Trissevgeni Stavrakou (jenny@aeronomie.be)

**Abstract.** Natural emissions (vegetation, soil, lightning) are the dominant sources of non-methane biogenic volatile organic compounds (BVOCs) and nitrogen oxides (NO$_x$ ≡ NO + NO$_2$) released into the atmosphere over Africa. BVOCs and NO$_x$
interact with each other and strongly impact their own chemical lifetimes and degradation pathways, in particular through their influence on hydroxyl radical levels. To account for this intricate interplay between NO$_x$ and VOCs, we design and apply a novel inversion setup aiming at the simultaneous optimisation of monthly VOC and NO$_x$ emissions in 2019 in a regional chemistry-transport model, based on TROPOMI HCHO and NO$_2$ satellite observations. The TROPOMI-based inversions suggest substantial underestimations of natural NO$_x$ and VOC emissions used as a priori in the model. The annual flux over
Africa is increased from 125 to 165 Tg yr$^{-1}$ for isoprene, and from 1.9 to 2.4 TgN yr$^{-1}$ and from 0.5 to 2.0 TgN yr$^{-1}$ for the soil and lightning NO emissions, respectively. Despite the NO$_x$ emission increase, evaluation against in situ NO$_2$ measurements at seven rural sites in Western Africa displays significant model underestimations after optimisation. The large increases in lightning emissions are supported by comparisons with TROPOMI cloud-sliced upper-tropospheric NO$_2$ volume mixing ratios, which even remain underestimated by the model after optimisation. Our study strongly supports the application of a bias
correction to the TROPOMI HCHO data and the use of a double-species constraint (vs single-species inversion), based on comparisons with isoprene columns retrieved from the Cross-track Infrared Sensor (CrIS).

## 1   Introduction

Tropical regions are an important source of natural fluxes of nitrogen oxides (NO$_x$ ≡ NO + NO$_2$) and biogenic volatile organic compounds (BVOCs), due to environmental conditions that favour their emissions. Both species are key drivers of tropospheric
chemistry through their impact on tropospheric ozone (Chameides et al., 1992) and hydroxyl radical (OH) levels through depletion or regeneration mechanisms (Fuchs et al., 2013; Hansen et al., 2017; Lelieveld et al., 2008), thereby altering the



oxidising capacity of the atmosphere and the lifetime of methane. In addition, BVOCs are important precursors of secondary organic aerosols (Carlton et al., 2009; Claeys et al., 2004) which, besides their direct effects on radiation, can also act as cloud condensation nuclei and alter cloud properties (Sporre et al., 2019). In recent years, air pollution in Africa has raised concerns

due to rapid population and economic growth with harmful environmental and health repercussions (Brauer et al., 2012; Mead et al., 2023; Petkova et al., 2013; Singh et al., 2020). Megacities such as Lagos and Kinshasa, oil exploitation (e.g. in Nigeria), coal mining and power generation in South Africa, biomass burning that is predominantly human-initiated, and aerosol dust (predominantly from the Sahara and Sahel) are key examples of sources affecting air quality over the continent. While anthropogenic emissions are the dominant source of $NO_x$ globally (Soulie et al., 2024), natural sources from soil and lightning

are responsible for ca. 30% of the total global $NO_x$ emissions into the atmosphere and contribute for even a greater share in remote environments (Jaeglé et al., 2005). Soil $NO_x$ emissions over Africa, due to soil microbial activity and fertiliser use, can be strong enough to be captured by spaceborne $NO_2$ observations (Jaeglé et al., 2004, 2005; Stavrakou et al., 2013), while Central Africa experiences amongst the largest lightning flash rates globally (Cecil et al., 2014). The evergreen tropical forests in West and Equatorial Africa are a major source of isoprene, by far the dominant BVOC emitted by vegetation (Guenther et

al., 2012; Lathière et al., 2006; Messina et al., 2016; Sindelarova et al., 2022). According to current estimates, ca. 80% of the total isoprene emission takes place in the tropics due to high temperature and solar radiation, and the widespread presence of broadleaved trees (Guenther et al., 2012; Opacka et al., 2021). Global bottom-up inventories for soil and lightning fluxes range from 1.3 to 6.6 TgN yr$^{-1}$ (Murray, 2016; Vinken et al., 2014; Weng et al., 2020), and from 4 to 34 TgN yr$^{-1}$ (Steinkamp and Lawrence, 2011; Yan et al., 2005), respectively, and isoprene fluxes from 300 to 800 Tg yr$^{-1}$ (Guenther et al., 2012; Sindelarova

et al., 2022). These large ranges are a measure of the uncertainty of natural emissions, which stem from the paucity of local observations and complex mechanisms driving the emissions fluxes.

Ground-based measurements in Africa have emerged from international efforts such as the INDAAF network (International Network to study Deposition and Atmospheric composition in Africa) for *in situ* measurements (Akpo et al., 2015; Ossohou et al., 2019; Ouafo-Leumbe et al., 2018), and the rapidly growing Pandonia global network (PGN) that monitors tropospheric

column data (https://www.pandonia-global-network.org/) or more recently, the Multi-AXis Differential Optical Absorption Spectroscopy (MAX-DOAS) observations of $NO_2$ and HCHO at Kinshasa (Yombo Phaka et al., 2023) and at Bujumbura (Gielen et al., 2017). With the exception of the INDAAF network, those sites are located in urban areas. A small number of field campaigns took place on the African continent such as the Dynamics-Aerosol-Chemistry-Cloud Interactions in West Africa campaign (DACCIWA 2016, Knippertz et al., 2017), the African Monsoon Multidisciplinary Analysis (AMMA 2006,

Lebel et al., 2011); Redelsperger et al., 2006), the Experiment for the Regional Sources and Sinks of Oxidants (EXPRESSO; Greenberg et al., 1999), as well as the Southern African Regional Science Initiative (SAFARI 2000, Swap et al., 2002). This is complemented by short-term monitoring of soil NO fluxes as reviewed in Liu et al. (2017), for instance.

By virtue of their global coverage and continuous monitoring, spaceborne data are great alternatives to studying air composition in Africa and are often used to constrain model emission estimates obtained with chemistry-transport models

(CTMs). Since the oxidation of BVOC emissions is the dominant source of formaldehyde (HCHO) over African rainforests



(De Smedt et al., 2008; Meyer-Arnek and Burrows, 2005) with HCHO being a high yield product of the oxidation of those BVOCs and having a short atmospheric lifetime of several hours, spaceborne HCHO columns constitute an excellent proxy of BVOC emissions (Barkley et al., 2013; Bauwens et al., 2016; Kaiser et al., 2018; Marais et al., 2014; Millet et al., 2008; Palmer et al., 2003; Stavrakou et al., 2009). Moreover, the use of satellite $NO_2$ data provides valuable information on the spatial distribution and magnitude of the natural sources of $NO_x$ over Africa. Several studies have inferred top-down isoprene or natural $NO_x$ emissions from satellite HCHO and $NO_2$ column observations, respectively, based on GOME, SCIAMACHY, GOME-2 or OMI (Bauwens et al., 2016; Jaeglé et al., 2005; Kaiser et al., 2018; Marais et al., 2012; Martin et al., 2003; Miyazaki et al., 2012; Palmer et al., 2003; Qu et al., 2020; Stavrakou et al., 2008). Global emission totals from top-down inventories range within 7.9-20.4 TgN yr$^{-1}$ for soil NO (Boersma et al., 2005; Jaeglé et al., 2005; Miyazaki et al., 2017; Stavrakou et al., 2013; Vinken et al., 2014), 3.3 – 7.5 TgN yr$^{-1}$ for lightning (Miyazaki et al., 2020; Stavrakou et al., 2013) and 272-445 Tg yr$^{-1}$ for isoprene (Bauwens et al., 2016; Müller et al., 2024). Regional inversions over Africa led to a reduction of about 20% in isoprene emissions (Bauwens et al., 2016; Marais et al., 2012, 2014) with respect to the a priori estimates obtained from the MEGAN bottom-up inventory (Guenther et al., 2012).

However, the top-down estimates bear errors owing to uncertainties in the satellite retrievals, the challenge of differentiating co-occurring sources, and uncertainties in the relevant chemical processes, in particular in the relatively low $NO_x$ conditions (typically 0.1-1ppb) which are typical in Africa (Marais et al., 2012). For instance, previous inverse modelling studies generally did not account for the existence of biases in spaceborne HCHO (De Smedt et al., 2021; Vigouroux et al., 2020; Zhu et al., 2020). The bias corrections were recently shown to enhance top-down isoprene emissions over source regions (Müller et al., 2024; Oomen et al., 2024). Moreover, the oxidation of isoprene and its associated HCHO formation are non-linearly dependent on $NO_x$ and isoprene levels, which therefore impact both the isoprene lifetime and the yield of formaldehyde (Atkinson, 2000; Wolfe et al., 2016). VOCs are oxidised through various photochemical pathways leading potentially to the production of carbon dioxide and water, with HCHO being an important intermediate in this oxidation chain. The first stage of the photochemical oxidation of a hydrocarbon by OH generally leads to the production of organic peroxy radicals ($RO_2$). Under high-$NO_x$ conditions, the peroxy radicals produced from isoprene oxidation react preferentially with NO, producing HCHO and other carbonyls, that are further oxidised and eventually yield additional HCHO (Atkinson, 2013). Under low $NO_x$ conditions, other reactions take place, leading to the formation of organic hydroperoxides (ROOH) and isomerisation products (Peeters et al., 2009), and furthermore, the lower OH levels associated with lower $NO_x$ levels slow down photochemistry and delay the secondary formation of HCHO (Barkley et al., 2013; Marais et al., 2012). The $HO_x$-$NO_x$-VOC relationship is a linchpin for the derivation of a top-down isoprene emission inventory from orbital HCHO column observations. Marais et al. (2012) estimated errors between 40 and 90% in the inferred isoprene emissions with the highest error under low-$NO_x$ conditions (<1 ppbv). Acknowledging the intricate interplay between $NO_x$ and VOCs through the $HO_x$-$NO_x$-VOC feedback, we design and apply a novel inversion setup relying on the concomitant use of HCHO and $NO_2$ satellite observations from the Tropospheric Ozone Monitoring Instrument (TROPOMI; De Smedt et al., 2018; van Geffen et al., 2022) mounted on the Sentinel-5P (S5P) satellite to simultaneously infer VOC and $NO_x$ emissions from the African continent. The aim of this study



is to better quantify natural emissions and to assess the impact of a double-species optimisation on those inferred emissions. For this purpose, the MAGRITTEv1.1 chemistry-transport model (Müller et al., 2019) is employed and simulations are conducted for 2019 at 0.5° resolution. The model accounts for up-to-date OH-recycling mechanisms in isoprene degradation (Müller et al., 2019). This study is the first to (i) derive simultaneously $NO_x$ and VOC emissions based on TROPOMI data and (ii) to account for biases in TROPOMI HCHO columns in the derivation of African emissions.

The paper is structured as follows. Section 2 describes the methodology applied, namely the modelling framework including the chemistry-transport model and the inversion method, the list of sensitivity inversions performed to assess the robustness of the results, the a priori emissions used in the model, the satellite products that are used to constrain VOC and $NO_x$ emissions, and the in situ and satellite data products that are used to evaluate the top-down fluxes; Sect. 3 provides a detailed discussion and evaluation of the results against independent datasets; finally, Sect. 4 presents the conclusions of this study.

## 2  Methodology

### 2.1  MAGRITTEv1.1 chemistry-transport model

Simulations of atmospheric composition are performed using a chemistry-transport model (CTM), the Model of Atmospheric composition at Global and Regional scales using Inversion Techniques for Trace Gas Emissions (MAGRITTEv1.1) (Müller et al., 2019), which is based on the global CTM IMAGES (Bauwens et al., 2016; Müller and Brasseur, 1995; Stavrakou et al., 115 2015). It calculates the concentrations of 170 compounds with a time step of 6 h and a spin-up time of 6 months. The model simulation covers the African domain delimited by 30°S-17°N and 18°W-56°E with a 0.5°×0.5° horizontal resolution, between the surface and the lower stratosphere with 40 hybrid sigma-pressure levels in the vertical.

In particular, the oxidation mechanism of isoprene implemented in MAGRITTE relies largely on the Leuven Isoprene Mechanism (Peeters et al., 2014), the Caltech oxidation mechanism (Wennberg et al., 2018) and the experimental investigation 120 of Berndt et al. (2019). A thorough description can be found in Müller et al. (2019). Regarding the impact of isoprene on $NO_x$ levels, the model accounts for organic nitrate formation in isoprene oxidation by OH and $NO_3$ (Kwan et al., 2012; Paulot et al., 2009; Wennberg et al., 2018), $NO_x$-recycling pathways such as photolysis (Müller et al., 2014) and $NO_x$ terminal losses through heterogeneous hydrolysis in aqueous aerosols (Romer et al., 2016) and dry deposition (Nguyen et al., 2015).

Meteorological fields are obtained from ERA-5 analyses (Hersbach et al., 2020). The model uses anthropogenic emissions of 125 $NO_x$, CO, $SO_2$, $NH_3$, and non-methane VOC species from the global CAMS-GLOB-ANTv5.3 inventory (Granier et al., 2019). The biomass burning inventory is provided by the Quick Fire Emissions Dataset (QFED) version 2.4 (Darmenov and da Silva, 2015; Pan et al., 2020) with emission factors from Andreae (2019) and are vertically distributed according to Sofiev et al. (2013). Natural emissions of isoprene, monoterpenes, and methylbutenol and natural sources of $NO_x$ are prescribed as detailed in Sect. 2.1.1. Biogenic emissions of acetaldehyde and ethanol are parameterised following Millet et al. (2010), biogenic





methanol emissions from the MEGANv2.1 inventory, and biogenic CO emissions are accounted for following Müller and
Stavrakou (2005).

The inverse modelling module and full (chemistry/transport) adjoint model were developed to perform tracer inversions of
reactive species for the optimisation of emissions, as described in Sect. 2.2. Bottom-up and top-down constraints used for the
optimisation are described as follows.

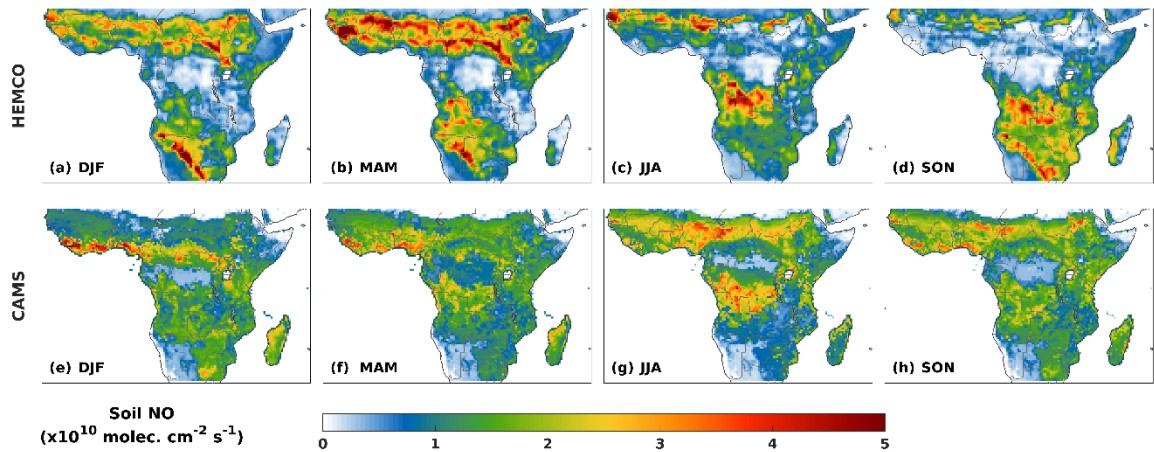

**Figure 1: Spatial seasonal distribution of above-canopy soil NO emissions (in $10^{10}$ molec.cm$^{-2}$s$^{-1}$) from HEMCO ((a)—(d))
and CAMS ((e)—(h)) inventory, for DJF (December-January-February), MAM (March-April-May), JJA (June-July-
August) and SON (September-October-November).**

## 2.1.1    Bottom-up inventories for natural NO$_x$ and VOC emissions

A priori soil NO emissions are obtained from the dataset of Weng et al. (2020) generated with the Harvard-NASA Harmonized
Emissions Component (HEMCO) version 2.1 model. Above-canopy fluxes accounting for biological and meteorological
drivers are calculated following Hudman et al. (2012). The latter includes (i) an exponential dependence on soil temperature
between 0°C and 30°C (constant at T > 30°C), (ii) a Poisson function scaling for soil moisture dependence, (iii) pulsing
according to the parameterisation of Yan et al. (2005), (iv) soil nitrogen input from chemical and manure fertilisers
(atmospheric N-deposition is not accounted for, as it contributes for only ca. 5% globally, according to Hudman et al. (2012)),
and (v) the canopy reduction factor. For this study, we use the monthly global emissions of soil NO$_x$ dataset based on
assimilated MERRA-2 meteorology, originally at resolution of 0.5° lat. × 0.625° lon., regridded to 0.5°× 0.5°. Averaged over
all years (1980-2017), the global total soil NO$_x$ emissions amount to 9.5 TgN yr$^{-1}$. As a sensitivity study, we use an alternative
dataset, the CAMS-BIO-SOILv2.4 inventory (Simpson and Darras, 2021), hereafter referred to as CAMS. This inventory is
also calculated following Hudman et al. (2012), with inputs from ERA5 reanalysis (Hersbach et al., 2020). While the global
total CAMS emission (9.1 TgN) is very similar to the HEMCO inventory, the spatial distribution and seasonal evolution
strongly differ, as shown in Fig. 1.





The lightning emissions of $NO_x$ in the reference case relies on the parameterisation of Price and Rind (1992) and Martin et al.
(2007) with convective cloud top heights derived from the ERA5 reanalysis (Hersbach et al., 2020), assuming a ratio of 0.75
between the NO production per flash due to intra-cloud and cloud-to-ground flashes (Martin et al., 2007). The monthly
averaged global distribution of lightning flash density follows the spaceborne climatology of the NASA Optical Transient
Detector (OTD) and Lightning Imaging Sensor (LIS) satellite instruments (Cecil et al., 2014). The lightning emissions are
distributed vertically according to Pickering et al. (1998). The mid-latitude lightning emissions are scaled up by a factor of 4,
following Martin et al. (2007), and the total lightning emission is scaled to 3.4 TgN yr$^{-1}$ globally. Seasonal distributions for
the reference lightning emissions are shown in Fig. 2.

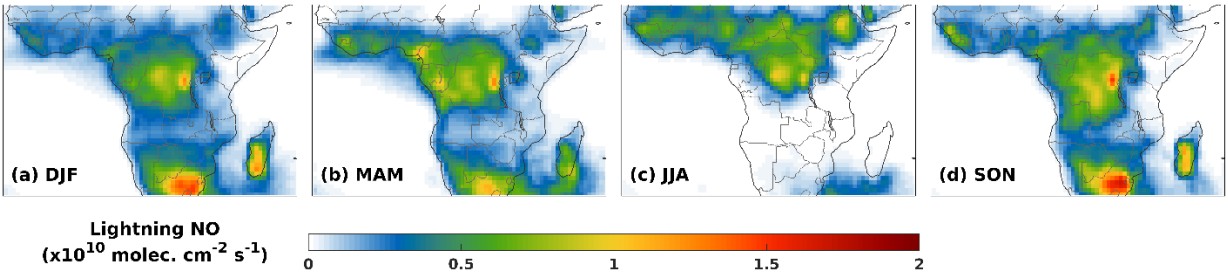

**Figure 2: Spatial seasonal distribution of lightning NO emissions (in $10^{10}$ molec.cm$^{-2}$s$^{-1}$) for the reference run ((a)—(d)) are displayed for DJF, MAM, JJA and SON.**

Biogenic emissions of isoprene, monoterpenes, and methylbutenol are calculated using the MEGANv2.1 model (Guenther et
al., 2012) coupled with the MOHYCAN multi-layer canopy model (Müller et al., 2008). Emissions are driven by
meteorological fields obtained from ERA5 reanalysis (Hersbach et al., 2020). The effect of soil moisture stress is neglected
($\gamma_{SM}$=1), since several studies (Bauwens et al., 2016; Opacka et al., 2022) indicated that the MEGANv2.1 parameterisation of
soil moisture dependence is very uncertain and needs improvement (Bauwens et al., 2016; Opacka et al., 2022; Jiang et al.
2018; Wang et al. 2023). Monthly LAI distributions at $0.5° \times 0.5°$ resolution (in m$^2$ m$^{-2}$) are based on the MODIS dataset
(MODIS15A2H collection 6). Our reference isoprene inventory, referred to as MEGAN, uses a gridded distribution of the
basal emission factor at standard conditions (provided in MEGANv2.1) based on species composition and species-specific
emission factors. As an alternative option, the ALBERI inventory (Opacka et al. 2021) will be used as a priori in the model.
This inventory uses constant emission factors per plant functional type (PFT) (Martin et al., 2007; Guenther et al., 2012) and
PFT distributions based on satellite-based datasets combining high resolution tree cover maps from the Global Forest Watch
(GFWv.1.9; Hansen et al., 2013) with MODIS land cover datasets (MCD12Q1; Friedl and Sulla-Menashe, 2019), as described
in Opacka et al. (2021).



The global annual emissions of isoprene in MEGAN and ALBERI in 2019 are estimated at 400 and 340 Tg, respectively, and their spatial distribution is shown in Fig. 3. The emissions account for the $CO_2$ response factor, parametrized following Possell and Hewitt (2011) and equal to 0.9092 for 2019 $CO_2$ concentrations

(https://gml.noaa.gov/webdata/ccgg/trends/co2/co2_annmean_mlo.txt).

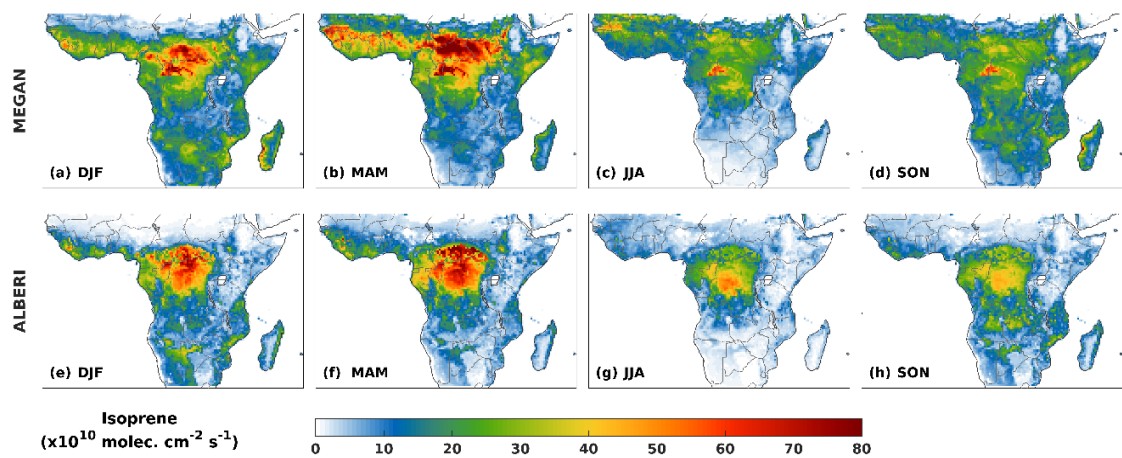

**Figure 3: Spatial seasonal distribution of isoprene emissions (in $10^{10}$ molec.cm$^{-2}$s$^{-1}$) from the MEGAN ((a)—(d)) and ALBERI ((e)—(h)) inventory are displayed for DJF, MAM, JJA and SON.**

### 2.1.2    Optimisation constraints: TROPOMI HCHO & NO₂ and INDAAF NO₂

We use tropospheric HCHO and $NO_2$ vertical column densities (VCD) measured by the Tropospheric Monitoring Instrument (TROPOMI) on board the ESA Sentinel-5 Precursor (S5P) satellite that flies on a low Earth polar orbit with an equatorial crossing time of ~13:30 local time. TROPOMI has a 2600 km wide swath and provides a near-daily global coverage at a spatial

resolution at $3.5 \times 5.5$ km$^2$ at nadir. The TROPOMI $NO_2$ dataset used in this study is the Products Algorithm Laboratory (PAL) version v2.3.1; for May 2018 to November 2021 this is the intermediate S5P-PAL reprocessing data (Eskes et al., 2021), which is followed by operational offline processing data (Eskes and Eichmann, 2024). These datasets include improved cloud pressure retrievals and a surface albedo correction, resulting in improved agreement with ground-based observations compared to previous product versions (van Geffen et al., 2022). Note that the v2.3.1 S5P-PAL data are no longer publicly available.

Both this data and the v2.3.1 operational offline data have been superseded by the v2.4.0 (re)processing data. Updates implemented for v2.4.0 (Eskes and Eichmann, 2024) include the use of an improved surface albedo database and reprocessed Level 1b input spectra with improved degradation correction. These updates may affect tropospheric $NO_2$ columns over polluted areas and vegetated areas, but as these effects are localised and relatively small, they are unlikely to significantly alter the main message of the present study. Tropospheric $NO_2$ data are regridded to 0.5° resolution, after selecting data with quality

assurance values (QA) above 0.75, per product recommendation. We further remove TROPOMI $NO_2$ data with strong biomass



burning (BB) influences. For that, given the large disparities in BB emissions among bottom-up inventories (Pan et al., 2020), we filter out the TROPOMI NO$_2$ data based on two global inventories, GFED4s (van der Werf et al., 2017) and QFED (Darmenov and da Silva, 2015), and consider the largest value between the two for each grid cell and month. More specifically, we exclude TROPOMI NO$_2$ data when either the daily BB flux, or the average flux from the previous 4 days, averaged spatially

within a 2.5°×2.5° box surrounding the 0.5°×0.5° TROPOMI pixel, exceeds 5×10$^{12}$ molec.cm$^{-2}$s$^{-1}$ (white areas on Fig. 4). This filtering reduces by 17% (from 8.5×10$^4$ to 7.0×10$^4$) the number of monthly 0.5°×0.5° TROPOMI NO$_2$ pixels used in the inversions. The BB NO$_X$ emissions are not optimised in order to limit the number of categories in our inversion setup. Instead, the ratio of BB NO$_X$ to BB VOC is assumed constant, and the BB NO$_X$ emissions are adjusted based on the changes inferred for BB VOC emissions. The assumption of a constant ratio between NO$_x$ and VOC emission factors is clearly an

approximation, but it is of little consequence for the other categories due to the NO$_2$ filtering and to the short lifetime of NO$_x$ (a few hours), which limits the spatial extent of the influence of fire emissions on the NO$_2$ distribution. In contrast with this, the influence of fires on HCHO is sustained downwind by the oxidation of emitted VOCs (Marais et al., 2012; Trentmann et al., 2003), which makes more difficult to filter out the influence of fires in TROPOMI HCHO data. Validation studies (Compernolle, 2024; van Geffen et al., 2022; Verhoelst et al., 2021) reported biases in TROPOMI NO$_2$ columns based on

ground-based MAX-DOAS stations. Most of these stations show median columns above 2×10$^{15}$ cm$^{-2}$, i.e. higher than the typically observed TROPOMI NO$_2$ levels in Africa. Exceptions are two Japanese sites, Fukue and Cape Hedo, for which very large TROPOMI overestimations (> +60%) were found (Compernolle, 2024). However, because these stations are maritime, and the vertical sensitivity TROPOMI and MAX-DOAS are very different (Oomen et al., 2024), these validation results should be considered with caution. Therefore, in absence of a robust validation study in low NO$_X$ areas typical of African NO$_2$ levels,

no bias correction was applied on TROPOMI NO$_2$ column densities.

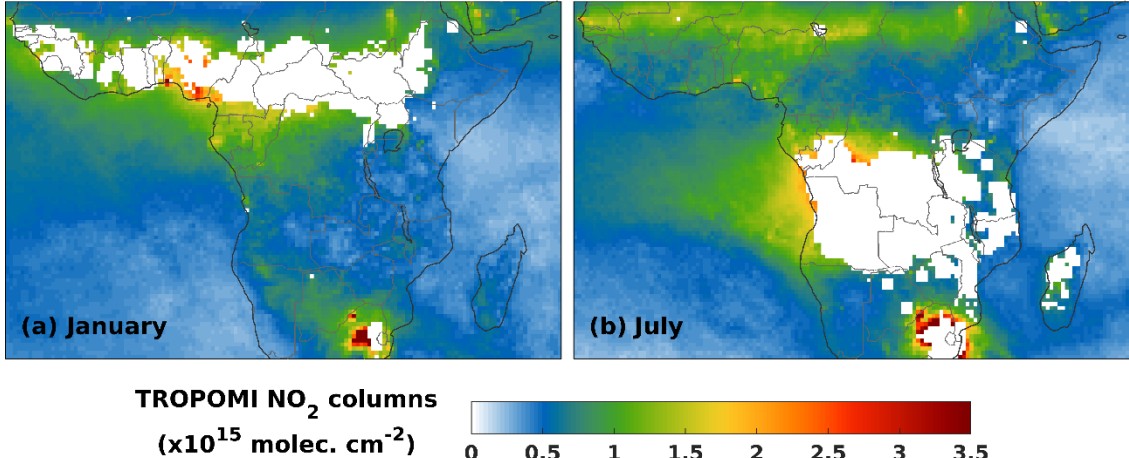

**Figure 4: Spatial distribution of TROPOMI NO$_2$ columns (in 10$^{15}$ molec.cm$^{-2}$) in (a) January and (b) July 2019 after the filtering of fire scenes.**



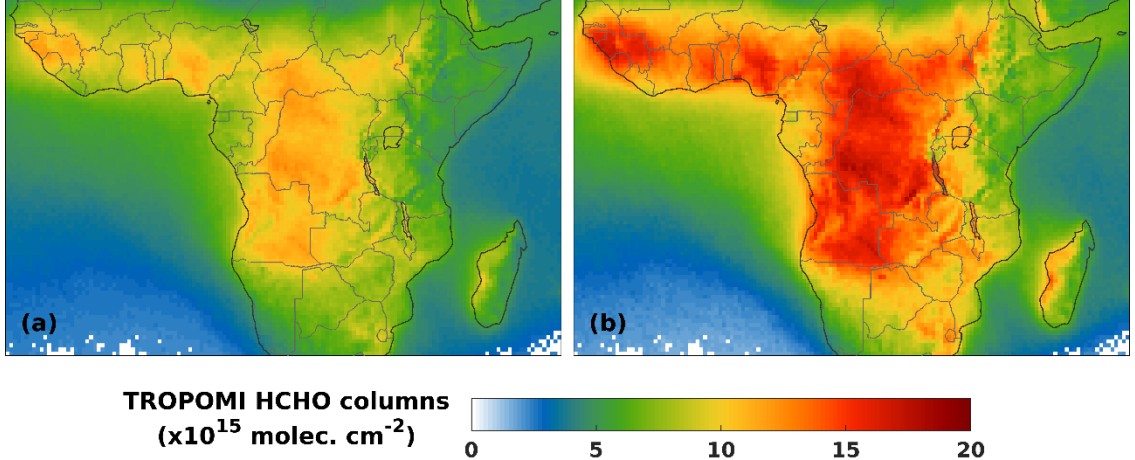

**Figure 5: Spatial distribution of annually averaged TROPOMI HCHO columns (in $10^{15}$ molec.cm$^{-2}$) in 2019 (a) before and (b) after the application of the bias correction.**

We use the Copernicus operational product TROPOMI S5p HCHO columns ($\Omega_{HCHO}$) (De Smedt et al., 2021) with QA>0.5
gridded onto the model resolution (0.5°) and corrected for bias using the following linear relationship based on Vigouroux et al. (2020):

$$\Omega_{HCHO}^{BC} = 1.587 \cdot \Omega_{HCHO} - 1.857 \times 10^{15} \text{ molec.cm}^{-2},$$

where $\Omega_{HCHO}^{BC}$ is the bias-corrected column. This formula is derived from a global-scale validation study of TROPOMI HCHO data using Fourier Transform Infrared (FTIR) data from 25 stations worldwide. The linear regression parameters are based on
monthly averaged FTIR data obtained within 2 hours of the satellite overpass and monthly TROPOMI HCHO columns averaged within a 20 km radius from the stations. Annually-averaged columns after bias correction are 30% higher on average, increasing from a domain-mean column of 6.63 to 8.58 $\times 10^{15}$ molec.cm$^{-2}$ (Fig. 5). The number of continental grid cells for TROPOMI HCHO in a year is about $8.5 \times 10^{4}$.

In a sensitivity study, an additional constraint on surface NO$_2$ mixing ratios obtained from a ground-based network of passive
samplers in Africa is applied on top of the constraint on TROPOMI HCHO and NO$_2$ data. The International Network to study Deposition and Atmospheric composition in Africa (INDAAF) network provides monthly surface concentrations of SO$_2$, HNO$_3$, NH$_3$, NO$_2$ and O$_3$. For the year 2019, we use measurements of monthly NO$_2$ concentrations at 7 INDAAF sites (Fig. 6) located in rural, unperturbed areas in West and Central Africa (Adon et al., 2010; Ossohou et al., 2023). We use a six-year record of observations (2015-2020) except for the Agoufou (2015-2018) and Bambey (2016-2020) sites for which only a
shorter period is available. Adon et al. (2010) reports that inter-comparison tests between passive and active samplers were performed in order to assess the performance and reliability of INDAAF passive instruments. A correlation of 0.95 within the 5-35 ppb range was inferred with a detection limit at 0.2±0.1 ppb and the reproducibility was found to be 9.8%. A linear regression yielded y = 0.9 x + 0.64 where y denotes the active samplers and x the passive method used by INDAAF.



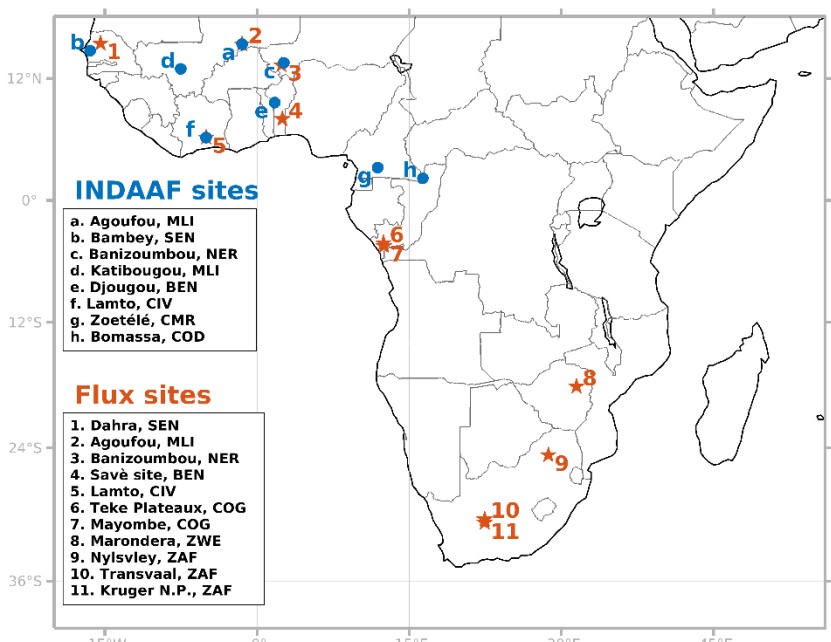

**Figure 6: Locations of in situ NO₂ concentrations sites from the INDAAF network (blue) providing NO₂ mixing ratios and in situ soil NO flux measurement sites (orange) used in this study.**

## 2.2 The inversion set up and sensitivity tests

Top-down (TD) emission estimates are obtained through an inversion approach that relies on the full adjoint model technique detailed in Müller and Stavrakou (2005). The MAGRITTEv1.1 model is coupled with an iterative minimisation algorithm optimising the emissions fluxes in order to minimize the discrepancies between the modelled and the observed satellite columns while accounting for errors in the a priori emissions and column observations (Fig. 7).

The a priori emission distributions for a given species are expressed as follows:

$$G_0(x,t) = \sum_{j=1}^{m} \Phi_j(x,t)$$

with $\Phi_j(x,t)$ the initial flux depending on spatial coordinates (latitude, longitude) and time (month) and $j=1... m$ the emission categories. The top-down fluxes $G(x,t)$ are expressed in terms of the a priori emission distributions $\Phi_j(x,t)$ and a dimensionless control parameter $\boldsymbol{f} =(f_j)$ as follows:

$$G(x,t) = \sum_{j=1}^{m} exp(f_j) \cdot \Phi_j(x,t).$$

The optimisation problem consists of the derivation of optimal dimensionless parameters $f_j(x,t)$ defined per emission category (biogenic, pyrogenic, anthropogenic, …), grid cell, and month. The method relies on the minimization of the scalar cost



function $J$, which includes two terms. The first term, denoted $J_{OBS}$, quantifies the discrepancy between the observations $\boldsymbol{y}$ and

the model predictions $H(\boldsymbol{f})$. The second term, denoted $J_B$, is a regularisation term that ensures that the optimised fluxes do not depart much from the a priori. The cost function is calculated as follows:

$$J(\boldsymbol{f}) = J_{OBS}(\boldsymbol{f}) + J_B(\boldsymbol{f}) = \frac{1}{2}[(H(\boldsymbol{f}) - \boldsymbol{y})^T \boldsymbol{E}^{-1}(H(\boldsymbol{f}) - \boldsymbol{y})^T + \boldsymbol{f}^T \boldsymbol{B}^{-1}\boldsymbol{f}]$$

where $\mathbf{E}$ and $\mathbf{B}$ are the error covariance matrices. The former is assumed diagonal and accounts for both observational errors and model representativeness errors. The $\mathbf{B}$ matrix accounts for errors on the a priori emission parameters as well as for

correlations between errors for different parameters. The adjoint of the model calculates the gradient of $J$ with respect to the parameter vector $\boldsymbol{f}$. The computation of the cost function and its gradient is performed iteratively until convergence is attained, that is, when the norm of the gradient of the cost function is decreased by a large factor (>20) with respect to its initial value, which typically requires 30-60 iterations.

For this study, a simultaneous inversion of VOC and $NO_x$ emissions is performed, using monthly averaged TROPOMI HCHO

and $NO_2$ column data as top-down constraints, such that the cost function contains four terms:

$$J(\boldsymbol{f}) = J_{OBS, \; VOC}(\boldsymbol{f}) + J_{B, \; VOC}(\boldsymbol{f}) + J_{OBS, \; NOX}(\boldsymbol{f}) + J_{B, \; NOX}(\boldsymbol{f})$$

In the joint inversion setup, six emission categories are optimised at 0.5°×0.5° resolution over the African domain (30°S-17°N; 18°W-56°E): anthropogenic VOC, biomass burning VOC, biogenic VOC, anthropogenic $NO_x$, soil $NO_x$ and lightning $NO_x$. Since fire scenes are filtered out of TROPOMI $NO_2$ data, the pyrogenic $NO_x$ emissions are not optimised, but are scaled as the

biomass burning VOC emissions. Fluxes are optimised only over land and when the a priori maximum monthly flux exceeds a given threshold depending on the category (see Table 1). This brings the number of control parameters to be optimised to about $3.8\times10^5$. Note that the total number of observational constraints is much lower, circa $1.6\times10^5$ after filtering. Filtered grid cells provide no constraints on $NO_X$ emissions for certain pixels and months. The effective number of control parameters is further reduced by defining spatio-temporal correlations between emission parameters of the error covariance matrix $\mathbf{B}$.


**Table 1: Emission categories considered for inversion and number of emission parameters per category. Emissions are optimised for a given category and pixel only when the maximum monthly flux exceeds the threshold given in the second column.**

| Categories | A priori flux threshold (molec.cm$^{-2}$s$^{-1}$) | Number of parameters |
|---|---|---|
| **Anthropogenic VOC** | $10^{11}$ | $7.10\times10^3$ |
| **Pyrogenic VOC** | $10^{10}$ | $57.8\times10^3$ |
| **Biogenic VOC** | $10^{11}$ | $72.9\times10^3$ |
| **Anthropogenic NO$_x$** | $5\times10^9$ | $20.2\times10^3$ |
| **Soil NO$_x$** | $10^8$ | $87.2\times10^3$ |
| **Lightning NO$_x$** | $10^8$ | $136\times10^3$ |



Spatial and temporal correlations are assumed to be decoupled, i.e. $B(r,t) = B(r).B(t)$, where $B(r)$ and $B(t)$ are the spatial and temporal components, respectively. The assumed error on the a priori fluxes for all emission parameters is taken equal to
1.1 (corresponding to an uncertainty factor of 3). The spatiotemporal correlation lengths of the biogenic and pyrogenic covariance matrices are set at 500 km for both VOC and $NO_x$. Acknowledging the arbitrary nature of these choices, sensitivity tests will be conducted to assess their impact (Sect. 2.3). Anthropogenic and biomass burning spatial and temporal correlation are defined as in Stavrakou and Müller (2006). For the soil and lightning emissions, the spatial correlations are assumed to decrease exponentially with the distance $d_{ij}$ between two grid cells $i$ and $j$, with a decorrelation length $l$ set to 500 km:

$$B_{ij}(r) = (\sigma_i/\phi_i)(\sigma_j/\phi_j)\exp(-d_{ij}/l),$$

where $\sigma_i/\phi_i$ is the relative error on the flux. The diagonal components of the error covariance matrix are the square of the relative errors on the fluxes emitted in grid cell $i$: $B_{ii}(r) = (\sigma_{\phi_i}/\phi_i)^2$. For isoprene, the error covariance matrix is defined based on the errors on the fluxes emitted from different grid cells and depends on the plant functional types (PFT) defined in the MEGAN-MOHYCAN model. The non-diagonal terms of the **B** matrix are defined as follows:

$$B_{ij}(r) = \sum_n e_i^n e_j^n \ (\sigma_i/\phi_i)\ (\sigma_j/\phi_j)\exp(-d_{ij}/l)$$

where $e_i^n$ represents the fraction of flux emitted in cell $i$ by the PFT $n$. In this way, the correlation is zero if the PFTs in pixels $i$ and $j$ are completely different. The temporal correlation of the errors is assumed constant for soil, set at 0.3. For the lightning and isoprene emissions, it is assumed to decrease linearly from 0.7 for consecutive months to 0.4 after 6 months. The error covariance matrix **E** is assumed to be diagonal, that is, errors are assumed to be uncorrelated. The errors on the monthly
TROPOMI averages are estimated from the reported retrieval errors, to which an assumed model/representativity error (2 $\times 10^{15}$ molec.cm$^{-2}$ for HCHO, 5 $\times 10^{14}$ molec.cm$^{-2}$ for $NO_2$) is quadratically added. In the case of HCHO, the reported error has systematic and random components, and the random part is strongly reduced upon data averaging. Regarding $NO_2$, an error correlation of 50% is assumed between retrieval errors on individual measurements contributing to the monthly average (Eskes et al., 2003; Stavrakou et al., 2013).

Monthly averages relying on less than 4 days of valid data and grid cells with less than 10 valid observations, as well as oceanic columns are omitted from the inversion. The modelled monthly averages are sampled at the time and location of the TROPOMI monthly means, and account for the averaging kernels of the observations. The simultaneous optimisation of VOC and $NO_x$ emissions takes advantage of the constraints provided from both satellite datasets, while accounting for $NO_x$-VOC chemical interdependencies. The impact of the two-species inversion will be assessed against the more usual single species inversions
(Sect. 2.3).





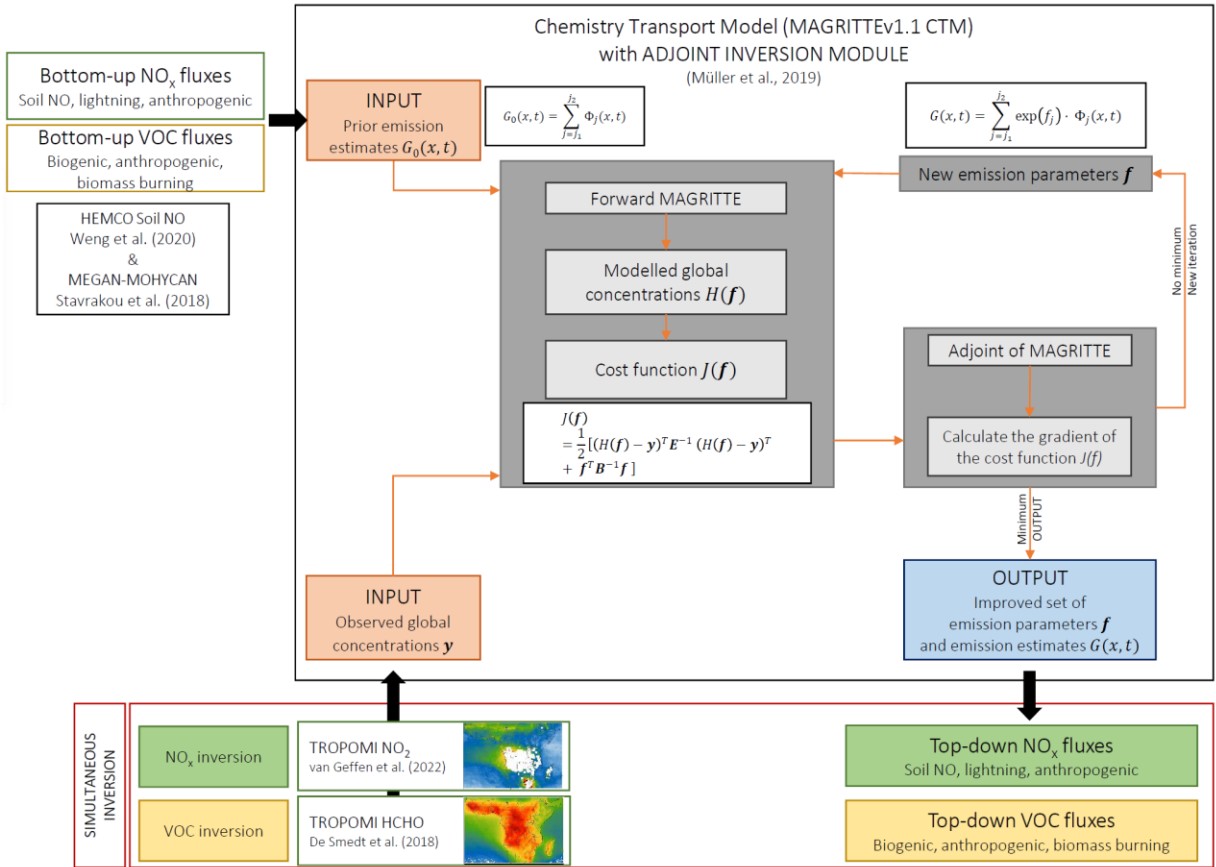

**Figure 7: Schematic flowchart of the inverse modelling approach using the adjoint inversion module of MAGRITTEv1.1 with inputs from bottom-up inventories and satellite observations. Description of the different annotations used in the flowchart can be found in Sect. 2.2.**

The reference inversion run, referred to as STD, is performed using a priori emissions for the natural sources of VOC and $NO_x$ provided by MEGAN, HEMCO and the lightning parameterisation of Price and Rind (1992), scaled globally to 3.4 TgN yr$^{-1}$.

For the evaluation of the range of uncertainty, we perform an ensemble of inversions (Table 2) aiming at assessing the impact of (i) single-species inversion (SVOC and SNOX), (ii) the bias correction on the HCHO satellite data (NOBC), (iii) the use of in situ $NO_2$ data from the INDAAF network (INDA) as additional constraints on $NO_x$ emissions, (iv) alternative choices for BVOC and soil emission inventories (ALBE and CAMN) and (v) different assumptions for critical inversion parameters (LERR and HERR). The a priori emissions used in simulations ALBE and CAMN were introduced and described in Sect.

2.1.1. All other inputs and model parameters are prescribed as defined in the STD simulation. For the INDA inversion, the assumed error on the INDAAF $NO_2$ mixing ratio constraint is set at 10%.





**Table 2: Description of the performed runs, including the standard inversion (STD) and the sensitivity runs.**

| Inversions | Description |
|---|---|
| **STD** | Two-species optimisation based on TROPOMI HCHO and $NO_2$ |
| **SVOC** | Single-compound inversion using TROPOMI HCHO; only VOC fluxes are optimised. |
| **SNOX** | Single-compound inversion using TROPOMI $NO_2$; only $NO_x$ fluxes are optimised. |
| **NOBC** | As STD, without bias correction on TROPOMI HCHO |
| **INDA** | As STD, use INDAAF surface $NO_2$ as additional constraint |
| **ALBE** | As STD, use the ALBERI isoprene emission inventory as a priori (Opacka et al., 2021) |
| **CAMN** | As STD, use the CAMS-BIO SOILv2.4 soil NO inventory (Darras et Simpson, 2021) as a priori |
| **LERR** | As STD, lower errors for VOC and $NO_x$ natural emissions (factor of 2 instead of 3) |
| **HERR** | As STD, higher errors for VOC and $NO_x$ natural emissions (factor of 4 instead of 3) |

## 2.3 Datasets for validation

### 2.3.1 In situ measurements

The a priori and top-down soil NO fluxes are evaluated against in situ soil NO measurements obtained from an extensive literature review. Measurements were performed in various biomes through various campaigns that took place during the 1991-2016 period. The compilation comprises 19 in situ soil NO flux data in Africa from 11 sites (Fig. 6). The mean fluxes over the wet and dry seasons are given in Table 3 for all sites. Furthermore, the a priori and top-down surface concentration of $NO_2$

will be evaluated against the INDAAF data record introduced in Sect. 2.1.2.



**Table 3: Compilation of in situ flux measurements of soil NO from the literature. The mean fluxes are given in $10^{10}$ molec.cm$^{-2}$ s$^{-1}$, with distinction between the dry (D) and wet (W) season. Also given is the type of biome at the measurement site: savannah (S), forest (F) or mixed (M). Notes: (a) average of forests and/or savannah ecosystems mean fluxes weighted by the number of individual chamber measurements, (b) arithmetic average over three unburned sites, (c) arithmetic average of uncorrected NO fluxes at dry control sites of Shabeni 6 and Kambeni 1, and (d) arithmetic average of measurements over grasslands, forests, fertilised and unfertilised crops and fallow fields.**

|    | Location | Lon. | Lat. | Mean Flux | Month | Season | Biome | Reference |
|----|----------|------|------|-----------|-------|--------|-------|-----------|
| 1  | Mayombe, Congo | 12.50°E | 4.50°S | 1.74 | 6 | W | F | Serça et al. (1994) |
|    |          |      |      | 0.78 | 7 | D |   |   |
|    |          |      |      | 0.50 | 2 | W |   |   |
| 2  | Teke Plateau, Congo | 12.45°E | 4.2°S | 0.03 | 4 | W | S | Serça et al. (1998) |
| 3  | Lamto, Ivory Coast | 5.03°W | 6.22°N | 1.02[a] | 1 | D | M | Le Roux et al. (1995) |
|    |          |      |      | 0.74[a] | 5 | W |   | Serça et al. (1998) |
| 4  | Nylsvley, South Africa | 22.42°E | 30.83°S | 2.98[a] | 3 | W | S | Serça et al. (1998) |
| 5  | Krüger N.P., South Africa | 22.42°E | 30.83°S | 0.97[b] | 10 | W | S | Parsons et al. (1996) |
|    |          |      |      | 2.13[b] | 11 | W |   |   |
|    |          |      |      | 1.01[b] | 12 | W |   |   |
|    |          |      |      | 3.17[a] | 8 | D |   | Serça et al. (1998) |
| 6  | Transvaal, South Africa | 22.42°E | 30.53°S | 1.99[c] | 9 | D | S | Levine et al. (1996) |
| 7  | Marondera, Zimbabwe | 31.74°E | 18.18°S | 4.85[d] | 10-12 | W | S | Meixner et al. (1997) |
| 8  | Savè, Benin | 2.49°E | 8.03°N | 2.06 | 6-7 | W | S | Pacifico et al. (2019) |
| 9  | Banizoumbou, Niger | 2.40°E | 13.30°N | 2.62 | 8 | W | S | Le Roux et al. (1995) |
| 10 | Agoufou, Mali | 1.48°W | 15.34°N | 2.88 | 7 | W | S | Delon et al. (2015) |
|    |          |      |      | 0.98 | 8 | W | S |   |
| 11 | Dahra, Senegal | 15.43°W | 15.40°N | 2.45 | 7 | W | S | Delon et al. (2017, 2019) |
|    |          |      |      | 1.72 | 11 | D |   |   |

### 2.3.2   UT NO₂

While there are multiple sources of $NO_x$ in the upper troposphere (UT) (lightning, aircraft, deep convective uplift of boundary layer pollution, and downwelling from the stratosphere), lightning NO emissions are the dominant direct upper-tropospheric $NO_x$ source (Lamarque et al., 1996; Murray, 2016; Schumann and Huntrieser, 2007). Satellite-based $NO_2$ volume mixing ratios (VMR) in five discrete layers in the troposphere from the surface to the tropopause were obtained via the cloud-slicing



technique applied to the total columns of $NO_2$ from TROPOMI (Horner et al., 2024). This technique retrieves the $NO_2$ VMR using the observed relationship between the cloud top pressure and the column above optically thick clouds developed by

Ziemke et al. (2001) for ozone in the free troposphere. Five layers of seasonal averaged $NO_2$ VMRs at 1°×1° between 2018 and 2022 are available and were retrieved from corrected TROPOMI total columns using cloud fraction and cloud-top height data from the FRESCO-wide algorithm (Eskes and Eichmann, 2023). After regridding to a finer spatial resolution of 0.5°×0.5°, the cloud-sliced UT $NO_2$ was obtained by averaging the two upper tropospheric layers, namely 320-180 and 450-320 hPa. Here, we use the cloud-sliced TROPOMI UT $NO_2$ VMRs to assess the a priori and top-down lightning emissions. This dataset

will be further referred to as cloud-sliced $NO_2$.

### 2.3.3 CrIS

Satellite-based isoprene column products are obtained from the Cross-track Infrared Sounder (CrIS), a Fourier transform spectrometer flying in a sun-synchronous orbit onboard the NASA/NOAA Suomi-NPP satellite launched in October 2011 (Wells et al., 2020, 2022). The instrument features a high spectral performance (0.625 cm$^{-1}$ resolution in the longwave IR 650-

1095 cm$^{-1}$), low noise (e.g., ~0.04 K at 900 cm$^{-1}$ and 280 K) and near-global coverage twice daily (0130 and 1330 local time) with the afternoon overpass corresponding with the peak of isoprene emissions. The isoprene retrieval algorithm relies on the hyperspectral range index (HRI) to quantify column abundances (Franco et al., 2018) retrieved from thermal infrared radiances at the $\nu_{27}$ and $\nu_{28}$ (from 890-900 cm$^{-1}$) absorption regions. Monthly distributions at 0.5°×0.625° resolution (regridded onto 0.5°) for year 2019 are used to evaluate the a priori and optimised isoprene columns.


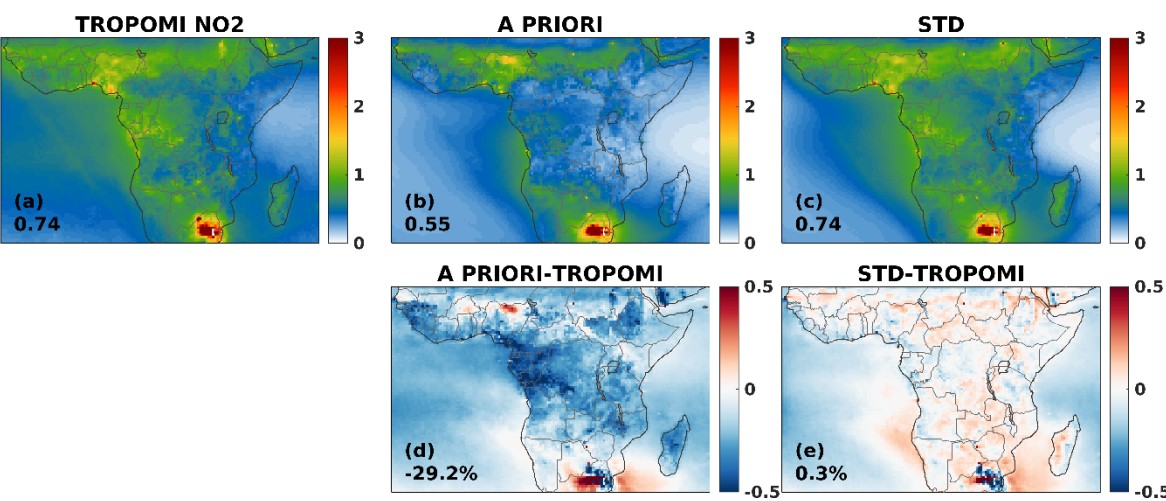

**Figure 8: Spatial distribution of annual $NO_2$ columns (in $10^{15}$ molec.cm$^{-2}$) of (a) TROPOMI, (b) a priori model and (c) the STD inversion. Average annual columns over land are provided inset. The absolute differences (in $10^{15}$ molec.cm$^{-2}$) are shown in the bottom row: (d) A PRIORI – TROPOMI and (e) STD-TROPOMI. The average percentage bias in $NO_2$ column densities over the**

**continent is provided inset.**



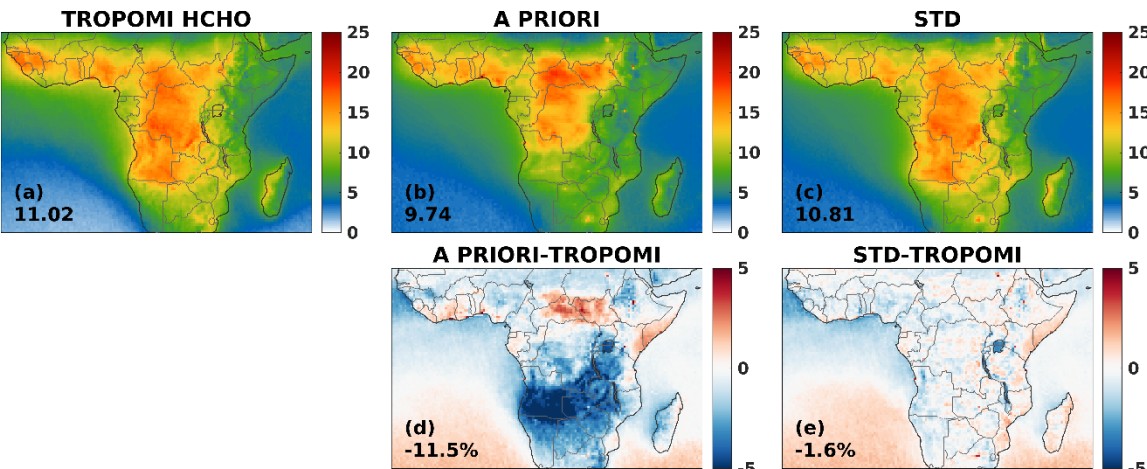

**Figure 9: As in Fig. 8, but for HCHO.**

# 3    Results and discussion

## 3.1  Optimised HCHO and NO₂ columns

As expected, the emission optimisation leads to a very good agreement with TROPOMI observations, both in terms of spatial distribution and magnitude (Fig. 8 and 9). The domain-averaged MAGRITTE $NO_2$ column over land increases from $0.55 \times 10^{15}$ molec.cm$^{-2}$ in the a priori run to $0.74 \times 10^{15}$ molec.cm$^{-2}$ in the STD inversion, resulting in a strong bias reduction from -26% to almost zero after inversion. The a priori model underestimation is ubiquitous in Africa and is most pronounced in Eastern Africa (-34%, region 2 in Fig. 10) and above the rainforest (-42%, region 3). These large biases are largely reduced after

inversion. Discrepancies still persist, however, in the South African Highveld region (Fig. 8), where the annual TROPOMI $NO_2$ columns reach their highest value over the domain (ca. $14 \times 10^{15}$ molec.cm$^{-2}$). This hotspot stems from the strong anthropogenic emission sources located in the Highveld plateau, from the densely populated cities of Johannesburg and Pretoria and from several unregulated large coal power plants and highly concentrated, energy-intensive industrial facilities in the Mpumalanga region (Matandirotya and Burger, 2021). Tropospheric column densities over the oceans are very low due to

the low emission over ocean and short $NO_x$ lifetime. The main sources of $NO_x$ over the ocean are in-cloud lightning and continental outflow, in addition to a small contribution from ship and aircraft emissions (IPCC, 2007). Moreover, TROPOMI retrieval errors are much higher over the ocean than over continents (readily above 100% and ranging up to 600%) and therefore, oceanic grid cells are not considered in the optimisation. Nevertheless, the bias over the ocean is also reduced after inversion, from -34% in the a priori to -16% in the STD run, with improvements occurring mainly along the west coast of

Africa due to the transport of continental air masses by the southeast trade winds (Fig. 8).

Likewise, the domain-averaged bias in HCHO columns is improved after inversion, from -12% to -2% (Fig. 9). The a posteriori columns match very well the bias-corrected HCHO data south of 8°S, where the large negative bias of the a priori simulation



(-28%) (regions 4 and 5 of Fig.10) is largely reduced, as a result of the strong enhancement of isoprene emissions, as discussed in Sect. 3.2. Over the equatorial region, the a priori columns reproduce very well the observed columns in the second half of
the year, and are moderately underestimated from February to May. The optimisation corrects this mismatch by inferring an increase of isoprene emissions. The a priori model overestimation over Central African Republic and South Sudan gives way to an excellent agreement after the inversion, which is realised by strongly reducing the biogenic and biomass burning emissions in these areas (Fig. 13 for isoprene and Fig. 14 for biomass burning).

For all regions of Fig. 10, the root-mean square error (RMSE) lies below $10^{15}$ molec.cm$^{-2}$ after optimisation for both species.
The a priori columns capture relatively well the observed seasonality and the a posteriori agreement is greatly improved. The performed sensitivity inversions lead to very similar results with respect to TROPOMI HCHO and NO$_2$ columns and are therefore not shown in Fig. 10 for the sake of simplicity.





**Figure 10: Time series of continental (a) NO₂ and (b) HCHO columns (in $10^{15}$ molec.cm⁻²) over the defined regions shown inset. TROPOMI NO₂ and bias-corrected TROPOMI HCHO columns (referred to as OBS) are depicted as blue diamonds with their corresponding retrieval errors. The a priori and optimised modelled columns are shown in black and red, respectively. The annual bias (in %) and the root-mean square error are shown inset for the a priori (black) and STD inversion (red). Regions are defined as follows: region 1 (4-12°N, 13°W-30°E), region 2 (8°S-12°N, 30-52°E), region 3 (8°S-4°N, 8-30°E), region 4 (8-18°S, 11.5-41°E) and region 5 (18-29.5°S, 11.5-36°E).**





## 3.2 Optimised fluxes

In the following we present the top-down distributions of soil and lightning $NO_x$ fluxes, isoprene and biomass burning VOC fluxes (Fig. 11-14, Fig. S1), summarized in Table 4 and Fig. 15. The satellite data indicate decreases in soil emissions in the Northern Hemisphere and a strong enhancement along the East coast of the continent (Fig. 11). Higher soil $NO_x$ emissions (up to a factor 4) are inferred by the STD optimisation in eastern Africa (between Tanzania and the Horn of Africa), moderate decreases (up to 30%) over the Congo rainforest, where a priori emissions are very low, and a decrease (up to 80%) in the largest part of northern Africa, in the Sahel region. Large differences in the optimised soil NO fluxes are inferred from the different inversions (SNOX, CAMN, INDA, Fig. 11) above the equatorial forest.

The single-species inversion SNOX leads to the same total annual soil NO flux as the STD inversion, 2.4 TgN (Fig. 15a). However, a stronger emission reduction is found by the SNOX inversion in Southern Congo and Angola compared to the base inversion (Fig. 11 f-g), whereas a soil emission increase is derived over Gabon, where a decrease was found by STD. This difference can be explained by the impact of isoprene emission changes on the lifetime of $NO_x$. Isoprene emissions have two opposing effects on the lifetime of $NO_x$. On one hand, the reaction of OH radicals with isoprene and its degradation products depletes OH levels, which tends to increase the lifetime of $NO_x$ (through the $NO_2+OH$ reaction); on the other hand, isoprene emissions enhance the sink of $NO_x$ due to the formation of organic nitrates ($RONO_2$) through $RO_2 + NO_2 \rightarrow RONO_2$, where $RO_2$ denotes peroxy radicals from isoprene oxidation (Romer Present et al., 2020). Organic nitrates are partially lost through either dry/wet deposition or heterogeneous reactions on aerosols, which result in a significant net loss of $NO_x$ (Müller et al., 2019). This effect tends to decrease the $NO_x$ lifetime. Over isoprene source areas with moderate to low $NO_x$ levels, the dominant influence is the increased loss of $NO_x$ through organic nitrates (Müller et al., 2019). Therefore, the higher (lower) isoprene fluxes derived by the STD inversion over Southern Africa (Gabon) (Fig. 13) lead to decreased (increased) $NO_x$ lifetimes, and therefore to increased (decreased) $NO_x$ emissions over these regions, in the STD run.

The INDA inversion leads to the largest increase in the equatorial region (by up to a factor of 6), as a consequence of using the INDAAF in situ $NO_2$ measurements at Zoetelé and Bomassa as constraints in the inversion, and to localised enhancements in the western regions coinciding with Katibougou, Lamto and Banizoumbou sites (Fig. 6). Unlike other inversions, CAMN predicts an enhancement in the soil NO emission in SH Africa (up to a factor of 4) with respect to the CAMS-BIO-SOIL bottom-up inventory, resulting in the highest annual flux estimate, 3.1 TgN (Fig. 15a), in very good agreement with a previous GOME-based estimate over Africa (3.3 TgN, Jaeglé et al. 2005), with almost equal contributions from northern and southern Africa.

The Sahel is of particular interest since it exhibits large pulses of soil NO emissions at the beginning of the rainy season in May-June and later on in September (Jaeglé et al., 2005). Based on OMI $NO_2$ columns, Vinken et al. (2014) found that this region emitted a total of 0.52 TgN in 2005, half of which occurred during the pulsing period from April to June. Here we derive a total of 0.37 TgN in 2019 with a primary peak in June, with 40% of the total top-down emissions released between April and June, and a secondary peak in October (0.04 TgN vs. 0.06 TgN in June).



Top-down lightning emissions in STD are increased by a factor of 4 in the STD case relevant to the a priori estimate (0.54 TgN) and range between 1.5 TgN in the CAMN run and about 2 TgN in the STD, SNOX and HERR inversions (Table 4 and
Fig. 15b). The highest fluxes are found in the equatorial region and in western Africa. Note that the inversion with the highest soil NO emissions (CAMN) has the lowest optimised lightning fluxes among all inversions. In all scenarios, TROPOMI suggests that NO emissions from lightning should increase everywhere (except over South Africa). The sum of soil and lightning $NO_x$ emissions inferred by the standard and sensitivity inversions is consistently close to ca. 4 TgN, 67% higher than the a priori estimates used in the inversion (2.4 TgN). The relative share of lightning emissions increases from 21% of the total
natural $NO_x$ flux in the a priori, to 45% in the standard inversion, at the expense of soil emissions. This relative share of lightning ranges between 30 and 50% across the sensitivity runs. This underscores the inherent difficulty to separate their relative contributions to the total column over Africa, where both sources are important.

The bias-corrected TROPOMI data indicate an increase of the annual isoprene fluxes, from 125 Tg in the a priori to 165 Tg in the benchmark STD inversion, whereas the range of the optimised emissions extends from 61 Tg for the NOBC inversion, for
which no bias correction is applied, to 185 Tg in SVOC, where only VOC fluxes are optimised (Fig. 15c and Table 4). The optimisation yields similar distributions of the scaling factor across the sensitivity inversions, except for NOBC that clearly stands out (Fig.13). In the STD inversion, the isoprene decrease north of the Equator occurs mainly in August with a substantial drop suggested by TROPOMI, while in the equatorial region, there is a moderate increase on annual basis from 29 to 36 Tg and a strong enhancement by 70% in isoprene emission in March (Table 4 and Fig. S1). Most prominently, the distributions
show strong emission enhancements in southern Africa, a factor of two or more over regions 4 and 5 (Table 4) and up to a factor of 5 locally. Similar emission adjustments, i.e. a decrease north of the Equator and an increase south of the Equator, were also derived from the inverse modelling of CrIS isoprene columns using the GEOS-Chem model, for year 2013 (Wells et al., 2020). The ALBE inversion leads to slightly lower annual isoprene emissions (135 Tg) compared to the base inversion estimate (165 Tg), which is due to the lower a priori total isoprene emissions of the ALBERI inventory (83 Tg, vs. 125 Tg in
the a priori of STD), due to the lower tree cover in this inventory (Opacka et al., 2021). At the lowest end of the range we find the inferred isoprene emissions of the NOBC inversion, which does not use a bias-correction for the TROPOMI HCHO data (61 Tg). This estimate is in close agreement with previous top-down estimates (60-70 Tg) based on OMI HCHO data not corrected for biases (Bauwens et al., 2016; Marais et al., 2012). However, OMI HCHO data were recently reported to be also subject to biases thanks to an extensive evaluation using FTIR and in situ aircraft data (Müller et al., 2024; Vigouroux et al.,
2020). After correcting for the bias in OMI HCHO columns, Müller et al. (2024) derived quite similar top-down isoprene emissions (153 Tg), as in the STD inversion using bias-corrected TROPOMI HCHO data (165 Tg), showing a good consistency between both top-down emission estimates, and much higher estimates with respect to previous studies. Finally, the inferred isoprene flux from the SVOC run is 14% higher than in the standard inversion (SVOC/STD: 185/165, Table 4). This is attributed to chemical feedbacks, namely the influence of $NO_x$ on OH levels and on the yield of HCHO from VOC oxidation.
More specifically, the overall higher top-down natural $NO_x$ emissions from the STD inversion (4 TgN, compared to 2.4 Tg in



the a priori), result in higher modelled HCHO, and therefore relatively lower isoprene fluxes are required to match the observations, compared to the SVOC case.

For sub-equatorial Africa, CrIS-derived isoprene emissions reported in Wells et al. (2020) indicate an emission hotspot over Angola, in very good consistency with the spatial distribution and magnitude found in our HCHO-based emissions (Fig. 13).

Although the top-down biomass burning VOC emissions (Fig 14 and Table 4) remain close to the a priori estimates, important changes are found in the spatial patterns. Relevant to the a priori inventory (QFED), fluxes are reduced by a factor of three over western Africa, Central African Republic, southern Congo and Angola, and are strongly enhanced in Mozambique, South Africa and Madagascar (factor of 4-6), whereas moderate increases are derived above Nigeria (less than factor of 2). The derived STD pyrogenic fluxes (33 TgVOC, Table 4) show comparable spatial distribution with OMI-based estimates in Africa

(Müller et al., 2024), but are by 25% lower on average. A closer agreement with Müller et al. (2024) is achieved in the SVOC setting, within 10%. The top-down anthropogenic $NO_x$ and VOC emissions (Fig. S2) do not change much relative to the a priori except in Nigeria where $NO_x$ emissions decrease by about 30% compared to CAMS. Anthropogenic VOC emissions decrease from 7 to 4 TgVOC in the STD inversion.

Overall, the optimisation based on both TROPOMI HCHO and $NO_2$ reveals an increase in natural fluxes compared to the

bottom-up inventories, by 26%, 263% and 30% for soil, lightning and isoprene emissions, respectively, and a small biomass burning decrease (Fig. 14 and Table 4). A good general consistency is found between the results of the different sensitivity inversions, although several inversions stand out, more specifically the NOBC, SVOC, ALBE, and CAMN inversions.

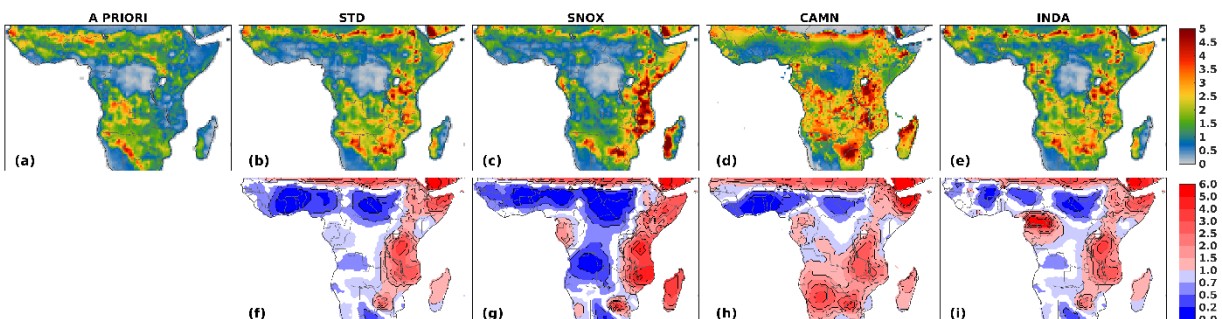

**Figure 11: The top row shows the spatial distribution of annual soil $NO_x$ emissions (in $10^{10}$ molec.cm$^{-2}$s$^{-1}$) of the a priori and resulting from the different inversions (STD, SNOX, CAMN and INDA). The bottom row shows the annual scaling factor resulting from the inversion (top row maps divided by the corresponding a priori). Blanks represent regions where the annual scaling factors lie between 0.9 and 1.1 i.e., 10% of change with respect to a priori fluxes.**



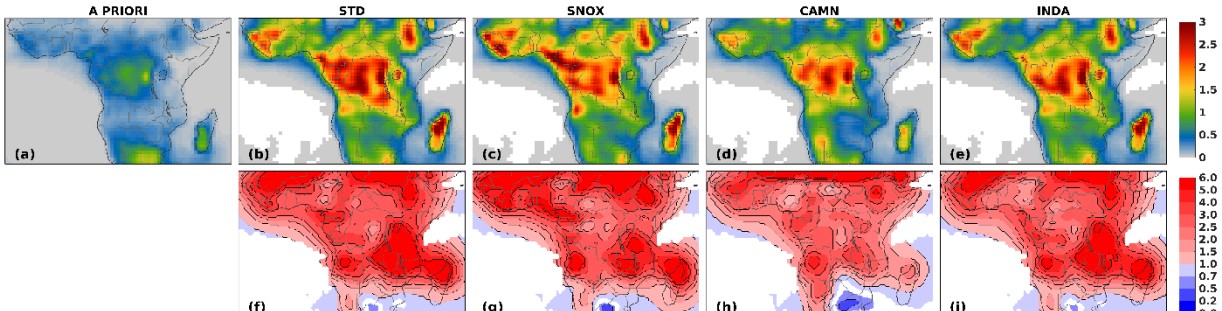

**Figure 12: As in Fig. 11, but for lightning NOx emissions.**

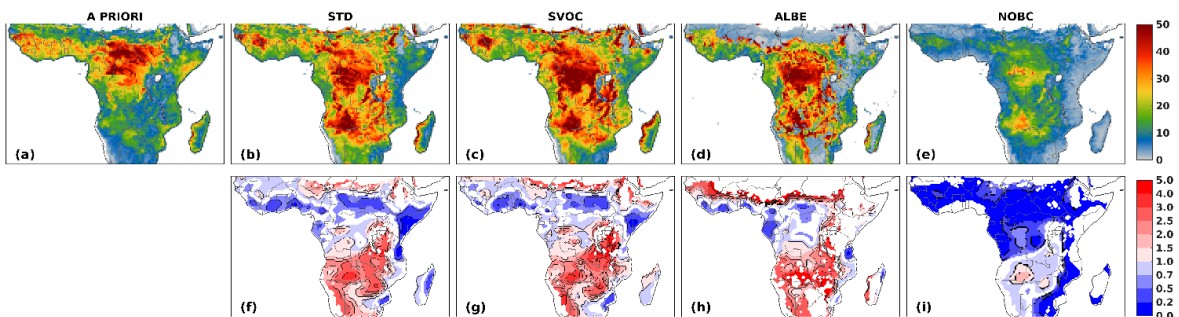

**Figure 13: The top row shows the spatial distribution of annual isoprene emissions (in $10^{10}$ molec.cm$^{-2}$s$^{-1}$) of the a priori and resulting from the different inversions (STD, SVOC, ALBE and NOBC). The bottom row shows the annual scaling factor resulting from the inversion (top row maps divided by the corresponding a priori). Blanks represent regions where the annual scaling factors lie**
**between 0.9 and 1.1 i.e., 10% of change with respect to a priori fluxes.**

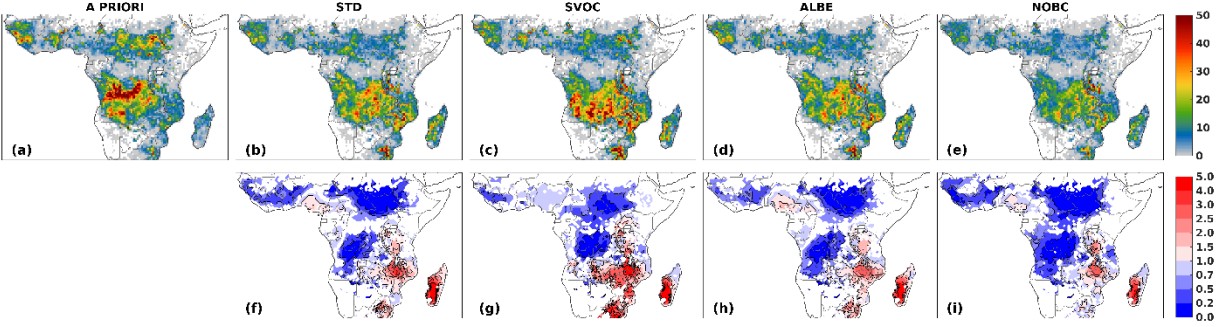

**Figure 14: As in Fig. 13, but for biomass burning VOC emissions.**






**Table 4: A priori and optimised emissions of soil and lightning NO (TgN), isoprene (Tg) and biomass burning VOC emissions (TgVOC) over the model domain and over the five regions depicted in Fig. 10. The ranges are based on the sensitivity inversions described in Table 2. Regions are defined as follows: region 1 (4-12°N, 13°W-30°E), region 2 (8°S-12°N, 30-52°E), region 3 (8°S-4°N, 8-30°E), region 4 (8-18°S, 11.5-41°E) and region 5 (18-29.5°S, 11.5-36°E).**


| Domain | Entire | Region 1 | Region 2 | Region 3 | Region 4 | Region 5 |
|---|---|---|---|---|---|---|
| Soil NO$_x$ (TgN) | | | | | | |
| A priori | 1.89 | 0.44 | 0.29 | 0.18 | 0.38 | 0.31 |
| STD | 2.39 | 0.26 | 0.53 | 0.19 | 0.54 | 0.38 |
| Sensitivity inversions | 2.17-3.09 | 0.24-0.42 | 0.44-0.63 | 0.18-0.44 | 0.49-0.73 | 0.37-0.42 |
| Lightning NO$_x$ (TgN) | | | | | | |
| A priori | 0.54 | 0.11 | 0.04 | 0.14 | 0.06 | 0.12 |
| STD | 1.96 | 0.45 | 0.22 | 0.54 | 0.31 | 0.18 |
| Sensitivity inversions | 1.51-2.06 | 0.35-0.52 | 0.15-0.23 | 0.44-0.56 | 0.20-0.33 | 0.16-0.19 |
| Isoprene (Tg) | | | | | | |
| A priori | 125 | 44 | 19 | 29 | 15 | 9 |
| STD | 165 | 39 | 21 | 36 | 36 | 20 |
| Sensitivity inversions | 63-188 | 15-42 | 7-27 | 16-43 | 16-39 | 8-21 |
| Biomass burning VOC (Tg VOC) | | | | | | |
| A priori | 36 | 9 | 4 | 8 | 14 | 2 |
| STD | 33 | 6 | 3 | 6 | 14 | 3 |
| Sensitivity inversions | 26-39 | 5-7 | 3-4 | 4-6 | 11-18 | 2-5 |



**Figure 15: Diagram of total annual emissions for all inversions defined in Table 2 of (a) soil NO$_x$ (in TgN), (b) lightning NO$_x$ (in TgN), (c) isoprene (in Tg isoprene) and (d) biomass burning VOC (in Tg VOC) emissions over the entire domain. A priori emissions are displayed in light blue and are followed by emissions resulting from inversions (dark blue) based on that bottom-up.**





## 3.3 Evaluation of results

Here, we present an evaluation of the optimisation against the INDAAF network of $NO_2$ mixing ratios (Sect. 3.3.1), in situ soil NO flux measurements (Sect. 3.3.2), satellite-based dataset of upper-troposphere $NO_2$ mixing ratio (Sect. 3.3.3) and isoprene columns of the spaceborne CrIS instrument (Sect. 3.3.4).

### 3.3.1 INDAAF $NO_2$ surface mixing ratios

Figure 16 displays monthly surface concentrations of $NO_2$ (in ppb) measured at the INDAAF sites in 2019 (Fig. 6), the average during 2015-2020, and the modelled concentrations from the *a priori*, the STD inversion and three sensitivity tests (SNOX, CAMN and INDA). The annual averages and the correlation with respect to 2019 measurements are provided inset, except for the Agoufou site for which no data were available in 2019.

The principal sources of near-surface $NO_x$ at the INDAAF sites are biomass burning and biogenic soil emissions, since they are all located in rural areas, except for Bambey and Djougou that have an urban influence. Overall, the base optimisation and sensitivity inversions do not depart much from the *a priori* except for the INDA case that shows a significant improvement as a result of the use of INDAAF data as additional constraints in the inversion. In this case, the observed and optimised concentrations for 2019 are in very close agreement, 1.0 vs. 0.9 ppb on average for all stations (Fig. 16), and the correlations are the highest among all simulations.

The highest $NO_2$ levels are observed in the dry savannas of Niger and Mali (Agoufou, Banizoumbou and Katibougou sites) that also exhibit a marked seasonal cycle with two peaks. The average 2015-2020 data show large standard deviation in Agoufou and most of the observed variability can be attributed to NO emissions from soils (Adon et al., 2010). Yet the *a priori* and the optimised $NO_2$ surface concentrations (STD, SNOX and CAMN) show very little monthly variability and the average mixing ratios are underestimated by a factor of 2-3 in comparison to the measured 2019 values in Banizoumbou and Katibougou. In Agoufou, the annual measured surface concentration (1.40 ppb) is three times higher than the model and a factor of 6 higher at the June peak (3 vs 0.5 ppb). The INDA inversion suggests that soil NO emissions should increase by a factor 3 and 1.5 on average in May-June-July compared to 2019 measurements at Banizoumbou and Katibougou, respectively. The second maximum occurring in October-November in Banizoumbou and Katibougou, most likely associated with the biomass burning season (Adon et al., 2010; Ossohou et al., 2019), is not captured by the standard optimisation which provides similar all-year-round $NO_2$ levels. By virtue of the extra constraints, the INDA inversion is able to capture more than 80% of the observed surface $NO_2$ variability at all sites. Discrepancies stem from the limited representativeness of one grid cell at 0.5° with respect to in situ measurements. Moreover, the large difference between the inversions based solely on satellite columns and the INDA inversion shows that the surface constraints help to better discriminate between soil and lightning $NO_x$ emissions. This is partly due to the lower sensitivity of satellite $NO_2$ retrieval to the boundary layer concentrations, in comparison to the middle- and upper troposphere. A similar model underestimation of $NO_2$ levels over the Agoufou,



Banizoumbou and Katibougou sites in 2005 was reported in the top-down study of Vinken et al. (2014), which was based on OMI $NO_2$ column data and the GEOS-Chem model.

In the wet savanna sites, Lamto and Djougou, the *a priori* and optimised $NO_2$ levels are in overall good agreement with the observations but with a moderate underestimation at Lamto (40%) in the standard inversion. Optimised $NO_2$ concentrations
from the INDA inversion at Lamto and Djougou agree very well with the observations, both in terms of magnitude and correlation. This match is realised by a doubling of the bottom-up soil NO fluxes over the entire year. Note however, that the observed $NO_2$ levels reported in Adon et al. (2010) are much higher at both sites. More specifically, the $NO_2$ concentration in Djougou reaches up to 3.5 ppb in the 1999-2007 dry season (Adon et al.,2010), more than twice the mean dry season level in 2019 and 2015-2020 (about 1.5 ppb, Fig. 16). This difference might be explained by the high average 1999-2007 fire emission
in Northern Hemisphere African savannas (430 TgC), compared to the 270 TgC in 2019 (van der Werf et al., 2017).

Finally, the Zoetelé and Bomassa forested sites exhibit a similar seasonality with maxima recorded in February-March, corresponding to the biomass burning season (Adon et al. 2010). On average, surface concentrations in the *a priori* and optimised model are largely underestimated (factor of 4 in Zoetelé, factor of 6 on Bomassa). CAMN leads to slightly higher annual averages in particular during the dry season compared to STD and SNOX inversions. According to the INDA results,
the soil NO fluxes should be increased by factors of 7 and 9 in Bomassa and Zoetelé, respectively, during the wet season (May-November) to match the observed surface $NO_2$ concentrations.





**Figure 16: Monthly surface volume mixing ratios of NO₂ (in ppb) for the 2019 measurements (blue diamonds), the average measurement for the 2015-2020 period (blue), and modelled concentrations from the a priori run (black) and from the inversions STD, SNOX, CAMN and INDA. Average volume mixing ratios (av. vmr, in ppb) and correlation are provided inset and are calculated with respect to INDAAF 2019 measurements. The vertical bars denote the standard deviation of the 2015-2020 measurements.**



### 3.3.2    In situ flux measurements

Figure 17 shows the average observed soil NO fluxes, the corresponding values from the STD and CAMN a priori runs and STD, INDA and CAMN inversions. The histogram displays fluxes in ascending order of site-averaged measured fluxes obtained with passive samplers at 11 different locations shown in Fig. 6 and reported in Table 3. Values of optimised fluxes for each corresponding measurement are provided in Table S1. The observations span a wide range of values, from $0.03 \times 10^{10}$ molec.cm$^{-2}$ s$^{-1}$ in Teké Plateau in Congo to $4.85 \times 10^{10}$ molec.cm$^{-2}$ s$^{-1}$ at Marondera in Zimbabwe. High temporal and spatial

variability is typical of the soil NO fluxes at the African continent (Serça et al., 1998). The low flux in the Teké Plateau wet savannah, north of Brazzaville, is recorded for a vegetation type (*Hyparrhenia*) that could inhibit nitrification (Serça et al., 1998). On the other hand, the high fluxes in Zimbabwe from October to December were recorded during the wet season in a mixed vegetation including grass, forest and crops. The average high value can be rationalised by the presence of a pulsing effect and to the application of fertilisers (Meixner et al., 1997).

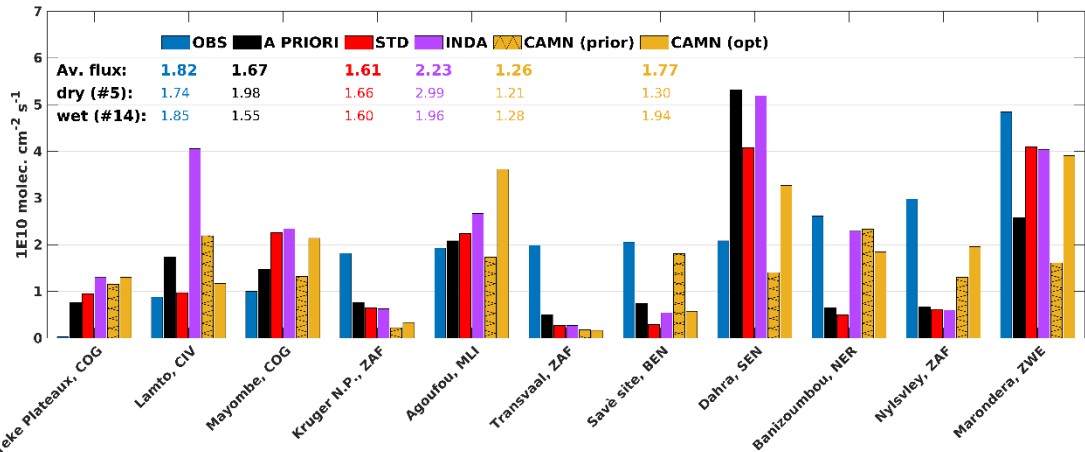


**Figure 17: Histogram of average soil NO flux measurements (in $10^{10}$ molec.cm$^{-2}$s$^{-1}$) from the compilation of Table 3 and Fig. 6. In situ observations are shown in blue, the a priori model in black and the inversion results in red (STD), magenta (INDA), and orange (CAMN). The a priori CAMN model is shown in hatched orange. The average fluxes (provided inset) are calculated based on the means for all data, over wet and dry seasons (cf. Table 3). The numbers in parentheses denote the number of available data.**


The a priori and optimised soil NO fluxes from the STD inversion show a slight overall underestimation of about 10% with respect to the observed all site-average of $1.82 \times 10^{10}$ molec.cm$^{-2}$ s$^{-1}$. Despite the relatively good overall agreement of optimised fluxes with respect to the measurements, we observe that the model tends to overestimate the low observed fluxes ($<1 \times 10^{10}$ molec.cm$^{-2}$ s$^{-1}$) and to underestimate the high ones ($>2 \times 10^{10}$ molec.cm$^{-2}$ s$^{-1}$), except at Dahra. Large discrepancies are found

at both Congo sites, Teké Plateau and Mayombe, where the inferred soil NO fluxes are strongly overpredicted. On average for all sites, the STD inversion yields a satisfactory agreement in both dry and wet seasons (Table S1), whereas the INDA results overpredict the observations, especially during the dry season (Table S1). All inversions, however, underestimate the fluxes in southern African sites, Kruger N.P, Transvaal, and Marondera, independently of the season.



The most recent data of this collection were obtained at the Savè site in Benin during the DACCIWA field campaign in June-July 2016 (Pacifico et al., 2019). The measured fluxes, $2.06\times10^{10}$ molec.cm$^{-2}$ s$^{-1}$ on average over the two months, were obtained during the wet season in a mixed ecosystem (bare soil, grassland, crop and forest) with the bare soil exhibiting the highest emissions. All inversions strongly underestimate these fluxes, by a factor of about 3.5 to 7, depending on the simulation. The best estimate at this site is provided by the bottom-up used in CAMN, while all inversions derive decreased emissions relative to the a priori flux.

The in situ flux sites of Agoufou, Banizoumbou and Lamto lie in the proximity of the corresponding INDAAF locations (Fig. 6). Comparisons at Agoufou indicate that the modelled soil NO fluxes in July and August agree well with in situ soil NO data, in contrast with the large model underestimation of the surface $NO_2$ measurements with respect to the 2015-2018 average, especially during the peak in July-August (factor of 6, Fig. 16). However, the significant variability in the surface $NO_2$ measurements, shown by the large standard deviation in Fig. 16, could partly account for this apparent inconsistency. At the Lamto site, the top-down fluxes provide a better agreement with in situ soil NO flux measurements, except for the INDA inversion which leads to a 5-fold enhancement of NO emissions despite showing the best agreement with INDAAF measurements (Fig. 16). At the Banizoumbou site, on the other hand, the strong soil NO increase (by a factor of 2.5) suggested by the INDA inversion is well corroborated by the observed soil NO fluxes in August (Fig. 17 and Table S1).

Several limitations in these comparisons should be acknowledged. First, the top-down estimates of soil NO fluxes are derived for 2019 whereas the in situ data were collected in the 1991-2016 period, which complicates the comparisons because of the high temporal variability of soil NO fluxes. Moreover, the spatial representativeness of the model at 0.5° horizontal resolution with respect to in situ measurements is another issue. An additional limitation lies in the comparison at forested sites because of the canopy uptake of $NO_2$. Indeed, in situ flux measurements are taken at the soil level whereas top-down inventories provide above-canopy fluxes that are generally lower than the above-soil fluxes. Bottom-up inventories account for this $NO_2$ uptake by applying the canopy reduction factor that is a highly uncertain parameter. Considering those shortcomings, the good performance of the optimised model regarding the average flux for all sites lends some confidence in the corresponding soil NO fluxes in Africa.

### 3.3.3 Upper-troposphere $NO_2$

Figure 18 shows seasonally-averaged mixing ratios of upper-tropospheric $NO_2$ mixing ratios between 180 and 450 hPa derived from TROPOMI $NO_2$ data averaged over two of the five layers 320-180 and 450-320 hPa and for 2018-2022, in comparison with the MAGRITTE a priori run and the STD inversion. According to the cloud-sliced $NO_2$ data, the average UT $NO_2$ mixing ratio throughout the year is about 50 pptv and is relatively constant from season to season with a maximal seasonal spread of 20%, ranging from 45 to 54 pptv. The inferred STD UT $NO_2$ mixing ratios display a larger seasonal variation (30-45 pptv), with a maximal seasonal difference of 70%. The optimisation strongly reduces the negative model bias with respect to the average values, from more than 70% in the a priori to ca. -35 to -17 % in the STD run, depending on the season.



The best model agreement is found in JJA when maximum values are attained. Regarding the spatial distribution, the optimised UT $NO_2$ distribution exhibits a stronger west-east gradient than the observations. Within the latitudinal band 15°S and 10°N, the negative bias after optimisation is significantly lower (<25%) in the western part than in the eastern part (~50%). This feature suggests either an underestimation of lightning NO emissions in the eastern part of the Congo Basin or in the vicinity of the Horn of Africa and/or a long-range transport of $NO_x$ from other regions, in particular oceans. Indeed, emissions over oceans are not optimised, and the LIS/OTD climatology used in the lightning parameterisation might underestimate the flash rate over oceans (He et al., 2022). The cloud-sliced TROPOMI $NO_2$ shows little interannual variability, therefore the comparison with $NO_2$ cloud-slicing dataset for 2019 is not expected to change the evaluation (Horner et al., 2024). The mismatch in modelled mixing ratios is similar in both layers (Fig. S3 and S4). Finally, using the earlier generation cloud-sliced UT $NO_2$ product from Marais et al. (2021), we find very similar results over Africa.

Despite the uncertainties and remaining biases, the above comparisons support the strong increase of lightning NO emissions suggested by the TROPOMI $NO_2$ data, and indicate that an even greater lightning source would be required to match the UT $NO_2$ measurements. In particular, strongly enhanced $NO_x$ emissions are suggested towards the eastern part of the continent, particularly over the eastern Congo Basin, where the a priori lightning source is low (Fig. 11 and 12). In situ flux or concentration data in eastern Africa would be needed to better constrain natural emissions of NO in this region.





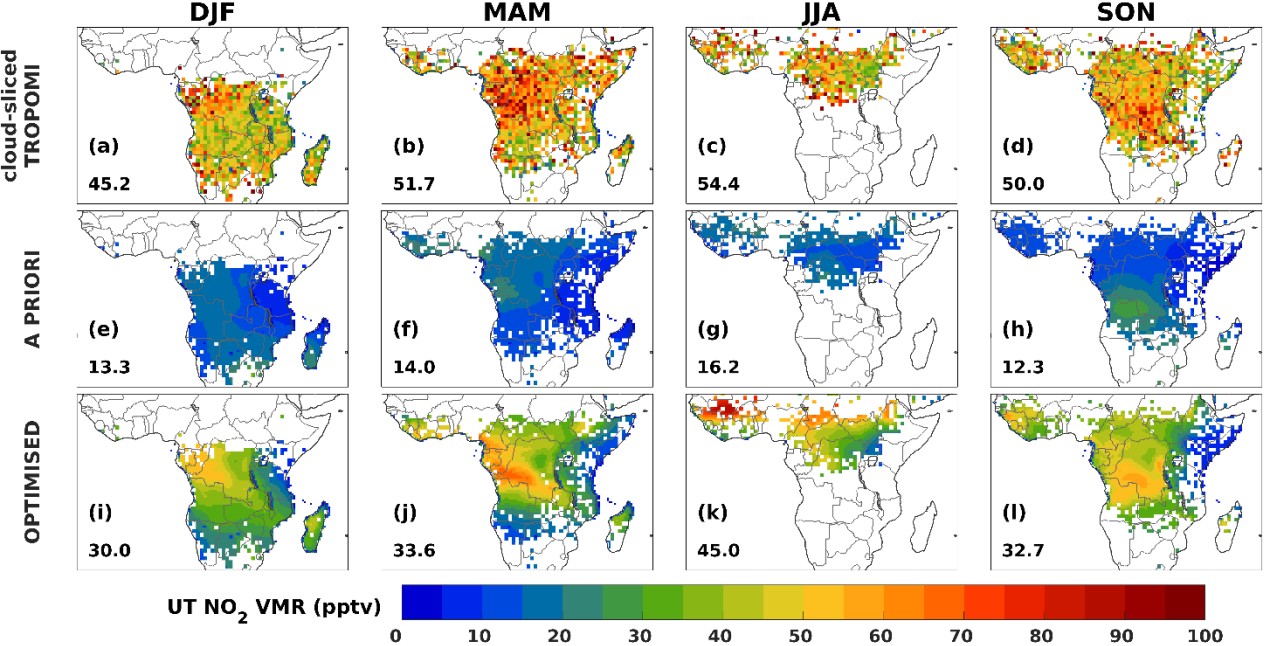

**Figure 18: Seasonal distributions of upper-troposphere NO₂ volume mixing ratios (in pptv) in December-January-February (DJF), March-April-May (MAM), June-July-August (JJA) and September-October-November (SON) for cloud-sliced TROPOMI NO₂ (Horner et al., 2024) (top row), the a priori (middle row) and after the STD inversion (bottom row). Blanks are grid cells without cloud-sliced retrievals due to lack of optically thick clouds over regions of subsidence.**

### 3.3.4 CrIS isoprene columns

The annually-averaged isoprene columns from CrIS show high values over forested areas in Congo (RDC), Angola and along

the coast of the Gulf of Guinea ($5\text{-}7\times10^{15}$ molec.cm$^{-2}$), and to a lesser extent along the Indian Ocean south of the Equator. The retrieved columns show low values over grassland, shrubs and bare soil regions, like in Namibia and the Horn of Africa, but the gradients are relatively weak over vast regions of moderate fluxes such as the Sahel. The spatial patterns of the a priori model are more contrasted, with a strong hotspot centred on the border between the two Congos ($> 15\times10^{15}$ molec.cm$^{-2}$), and much lower columns over much of the rest of Africa ($< 3\times10^{15}$ molec.cm$^{-2}$). The optimisation based on HCHO and NO₂ data

(STD) leads to a much improved agreement with CrIS, as shown by the bias reduction in annually-averaged columns over the entire domain, from -33% in the a priori to -10% after inversion, and also seen in the a posteriori spatial distribution of isoprene columns (Fig. 19). The TROPOMI observations suggest a displacement of the isoprene hotspot towards the south, and a strong increase of the overall column levels, in very good agreement with the CrIS data and in accordance with CrIS-based inversion results reported in Wells et al. (2020). The reduction of the isoprene fluxes over the Central African Republic and the

emergence of high isoprene levels over Angola, Zambia and southern Congo (RDC) after inversion is fully corroborated by the CrIS data lending good confidence to the top-down estimate. The isoprene fluxes derived here from TROPOMI in 2019





and inferred from CrIS in 2013 agree reasonably well with regards to the spatial distribution and scalar increments (Fig. 13, Fig. S13 in Wells et al. (2020)), as well as in terms of isoprene emission flux. Indeed, the CrIS-derived isoprene flux for January and April 2013, estimated at 10.9 and 9 TgC, respectively (Wells et al., 2020), is in line with the corresponding

TROPOMI-based flux from the STD inversion at 12.2 and 12.4 TgC for 2019 (Fig. S5). The higher values obtained from TROPOMI could be partly explained by the natural interannual variability between 2013 and 2019, given that 2019 was among the warmest years for Africa in the last century (World Meteorological Organization, 2020).

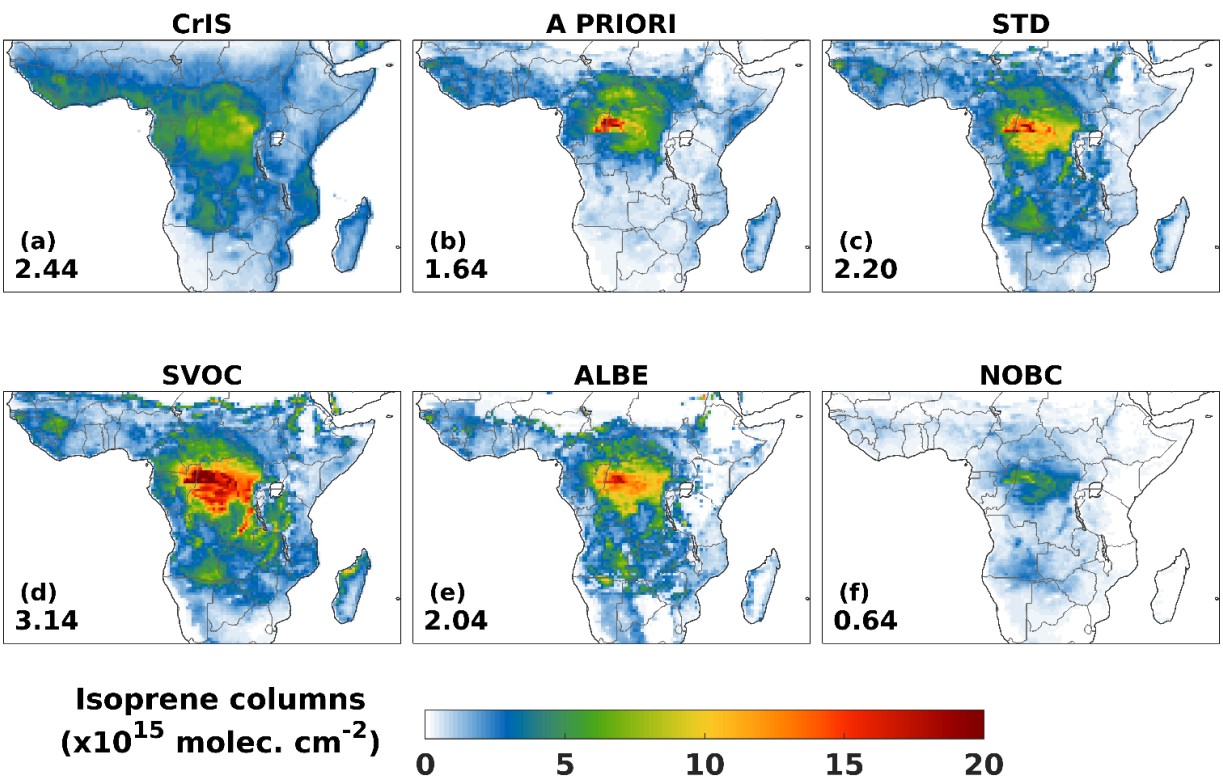

**Figure 19: Spatial distribution of annual isoprene columns (in $10^{15}$ molec.cm$^{-2}$) from (a) a CrIS-based product, and (b)-(f) modelled in the a priori and inversions (STD, SVOC, ALBE and NOBC). Average column over the entire continental domain is provided inset (in $10^{15}$ molec.cm$^{-2}$).**

The single-compound SVOC inversion, which is only constrained by HCHO data, worsens the agreement with CrIS and results in an increase by 42% of the domain-wide isoprene columns compared to the STD case (Fig. 19). This is explained by the

higher $NO_x$ emissions derived in STD compared to the a priori, leading to higher OH levels and therefore shorter isoprene lifetimes. Higher OH concentrations also promote HCHO formation from the oxidation of many precursor VOCs including methane and isoprene (Marais et al., 2012; Wolfe et al., 2016), therefore leading to overall lower isoprene emissions (Sect. 3.2) and columns. The ALBE inversion yields similar results, but the inferred domain-averaged columns are lower than with



STD, as a result of the lower top-down isoprene emissions derived by this inversion (135 Tg, 165 Tg in STD). Not applying a
bias correction in TROPOMI HCHO columns in the NOBC inversion results in very low isoprene abundances, by about 4
times lower than the CrIS yearly average over the studied domain, thereby providing strong evidence for the need to account
for biases in satellite HCHO data in the inversions. Note that the NOBC results are in line with past studies performed with
OMI that found that bottom-up African isoprene emissions (Sect. 3.2) were too high (Bauwens et al., 2016; Marais et al., 2012,
2014), but this is disproved by the CrIS isoprene data and by the strong evidence of biases in OMI and TROPOMI HCHO
data. It should be mentioned that the CrIS product does not include averaging kernels, which makes it impossible to account
for the scene-specific of CrIS vertical sensitivity in our comparisons, possibly leading to errors. The results obtained from the
inversions assuming either lower (LERR) or higher (HERR) errors in the a priori emissions lie within 4% of the STD estimated
domain-wide isoprene columns (2.16-2.28×$10^{15}$ molec.cm$^{-2}$) and exhibit similar distributions (not shown). The monthly cycle
of domain-wide CrIS data indicates weak variability, with a peak-to-trough difference of ca. 2, and two peaks in March-April
and in October, whereas the observed cycle shows much stronger amplitude over smaller regions, like Western, Eastern and
SH Africa (factors of 5-8, Fig. 20). The top-down STD columns overpredict the March-April peak in equatorial Africa (region
3) and fall short of capturing the observed seasonality in regions 1 and 2, leading to negative correlations after inversion. The
TROPOMI data suggest a column decrease relative to the a priori in the second half of the year, which is contradicted by the
gradual increase of CrIS isoprene columns from June to October and sudden drop in November and December. The a priori
model driven by the MEGAN inventory also fails to reproduce the CrIS seasonality in these two regions. The reasons for this
discrepancy are unclear. They could be related to a misrepresentation of OH levels in the model, since OH exerts a major
control on isoprene abundance, or to missing seasonal processes in the a priori model. While seasonal leaf area index (LAI)
may be fairly constant in the tropical forest, the different tree species tend to flush their leaves in different seasons which
means the fraction of high emitting LAI can vary and thus cause a seasonal cycle in isoprene emissions (Alves et al., 2023).
In equatorial Africa (region 3), the observed maximum in March-April reaching 5×$10^{15}$ molec.cm$^{-2}$ is overpredicted by a factor
of 2, but a satisfactory agreement between the optimised model and the observations is found for the rest of the year. In SH
Africa (regions 4 and 5), the optimised columns reproduce well the observed seasonal cycle, which exhibits a minimum during
the SH winter (June-August), and a maximum in summertime. Among all inversions, the largest model underestimation of
CrIS columns is found in the inversion assuming no bias correction on the TROPOMI HCHO data.



**Figure 20: Time series of monthly isoprene columns over the entire domain (a) and the five regions shown in Fig. 10. Isoprene columns from CrIS are shown in blue. The a priori, STD, SVOC, ALBE and NOBC modelled columns are in black, red, orange, green and magenta, respectively. Units are $10^{15}$ molec.cm$^{-2}$.**




## 4    Conclusion

The present study presents monthly top-down emissions of $NO_x$ and VOC at 0.5° resolution over Africa for the year 2019, derived from spaceborne HCHO and $NO_2$ tropospheric column retrievals and an adjoint-based inverse modelling framework built on the chemistry-transport model MAGRITTEv1.1. The main distinctive feature of the proposed joint inversion scheme

is the simultaneous optimisation of the VOC and $NO_x$ sources, taking their chemical interactions into account. The aim of this joint inversion setup is to achieve improved emissions through a better simulation of OH levels and VOC degradation pathways that are relevant to HCHO production and $NO_x$ sinks processes. Additional simulations were performed to assess the robustness of the results. Overall, our study suggests that the natural emissions of $NO_x$ and reactive VOCs (i.e. primarily isoprene) are underestimated by the inventories used as a priori in the simulations, although there are large seasonal and geographical

variations in the magnitude of the biases. More precisely, our standard, two-species inversion leads to the following conclusions.

- Soil emissions increase by 26% over the model domain, from 1.9 to 2.4 TgN $yr^{-1}$. The distribution of the soil NO scaling factor displays complex patterns of increases and decreases. Comparison with in situ flux measurements show

that on average, both a priori and optimised fluxes slightly underestimate the measurements, by about 10%. Despite this good overall agreement, the model tends to overestimate the low observed fluxes ($<1\times10^{10}$ molec.$cm^{-2}s^{-1}$), e.g. at the Congo sites, and to underestimate the high fluxes ($>2\times10^{10}$ molec.$cm^{-2}s^{-1}$), such as at the southern African sites.

- Lightning emissions are enhanced by a factor of almost 4 (+263%), from 0.54 to 2 TgN $yr^{-1}$. The increase is relatively

ubiquitous, and is corroborated by comparisons with cloud-sliced UT $NO_2$. An even greater lightning source would be required to remove completely the low bias with UT $NO_2$ data, particularly towards the eastern part of the continent. A large underestimation (~50%) of modelled UT $NO_2$ is found in the Horn of Africa, for reasons unclear. Either lightning emission or long-range transport of $NO_x$ from other regions could be too low in the model. In the vicinity of the Horn of Africa, the inversion leads to large changes in soil emissions (up to a factor of 4), but ground

measurements are lacking to corroborate this result.

- Chemical interactions between VOCs (primarily isoprene) and $NO_X$ play a significant role in the optimisation of isoprene and $NO_X$ emissions in Africa, as shown by the disparity between two-species vs single-species inversions. The general increase in $NO_x$ emissions suggested by TROPOMI $NO_2$ enhances HCHO formation from $CH_4$ and

isoprene, resulting in generally lower isoprene top-down emissions in the two-species inversion, compared to SVOC, the single-species run. The impact of isoprene emission changes on top-down $NO_x$ emissions is more complex because the two main sinks of $NO_x$, the $NO_2$+OH reaction and organic nitrate formation, have opposite responses to isoprene





fluxes, as isoprene depletes OH and favors organic nitrates. The net effect of isoprene changes is therefore dependent on the relative shares of the two sink processes to the total $NO_x$ loss.


- All inversions, except SVOC - the single species inversion based on TROPOMI HCHO - and NOBC - the inversion without bias correction based on TROPOMI HCHO - indicate that the bottom-up isoprene emissions from MEGAN2.1 should be increased by ~30% over the whole region, from 125 Tg to 165 Tg. The single-species inversion (SVOC) and NOBC lead to different estimates, 188 Tg and 63 Tg, respectively. The lower estimate of the NOBC inversion is a direct consequence of the lower TROPOMI HCHO columns over Africa (~25%) when correction is not applied; this result is consistent with previous top-down emission estimates over Africa. However, both inversions SVOC and NOBC lead to poorer comparisons with the isoprene column data retrieved from the CrIS, in terms of mean column density and spatial distribution. In particular, the NOBC run leads to a large underestimation of isoprene columns (~70%) compared to CrIS, whereas SVOC suggests a strong enhancement of isoprene columns over equatorial Africa that is not supported by satellite data. Furthermore, the decrease in isoprene fluxes over the Central African Republic, the strong increase in the southern regions and the emergence of high isoprene levels over Angola, Zambia and southern Congo after inversion are fully corroborated by the CrIS data, lending good confidence in the top-down estimate. The improved performance of the double-species inversion compared to SVOC supports the need to account for satellite $NO_2$ data through joint $NO_2$-HCHO inversions, especially in tropical regions where the distribution and strength of $NO_x$ sources is quite uncertain. However, we found that the seasonality of the isoprene columns is not well reproduced over North Africa, for reasons unknown. The comparison is made uncertain by several factors, such as the lack of averaging kernels in the CrIS data, which makes it impossible to account for the scene-specific vertical sensitivity of CrIS in the model comparison.

This study has a number of limitations. A major issue is the relative scarcity of field-based observations (in situ, ground-based or airborne data) that could provide us with a more comprehensive database for model evaluation and a better understanding of ecosystem-scale processes. There is a relatively extensive network of field stations across the African continent focused on the study and monitoring of greenhouse gases (López-Ballesteros, 2018), and further development of this infrastructure is foreseen with the installation of a low-resolution FTIR instrument near the CongoFlux tower (Sibret et al., 2022) in the

Yangambi Biosphere Reserve (0°48'52.5"N, 24°30'08.8"E) (https://icos-be.aeronomie.be/index.php/project). In addition, the CongoFlux tower could be an excellent platform to set up a VOC and $NO_x$ study in the heart of the tropical rainforest of the Congo Basin. Sites equipped with MAX-DOAS instrumentation in Kinshasa and Bujumbura (Gielen et al., 2017; Yombo Phaka et al., 2023), as well as the two PGN stations, Potchefstroom and Wakkerstroom in South Africa, are located in urban areas and are less suitable for the study of natural emissions. Furthermore, there is currently a gap in field-based data in East

Africa, which is among the most sparsely observed regions in the world (Bililign et al., 2024). The limitations of our study are also related to the model. The parameterisation of lightning emissions in the model could be improved by exploring new



lightning schemes (Luhar et al., 2021; He et al., 2022). The photolysis of particulate nitrate, a recycling path of $NO_X$, should be implemented, as it would increase the $NO_X$ lifetime and therefore the $NO_2$ columns (Shah et al., 2024; Ye et al., 2016). In addition, the recent evidence of water vapor absorption in the UV (290 to 350 nm) could have important implications,

particularly in tropical environments due to its potential impact on the retrieval of HCHO columns and on the formaldehyde lifetime and OH levels in chemistry-transport models (Pei et al., 2019; Prather and Zhu, 2024). The benefits of the double species inversion for deriving improved VOC and $NO_x$ fluxes will be further investigated in other regions such as South America, where the reported large underestimation of natural nitrogen oxide emissions (Lee et al., 2024; Wells et al., 2020) might have a substantial impact on the inversion of VOC fluxes.


**Author contribution**

BO carried out the analysis and wrote the manuscript. TS and JFM designed the inversion scheme and performed the optimizations. IDS provided the TROPOMI HCHO retrievals and advice for their use. JVG developed the TROPOMI $NO_2$

product version and provided recommendations for its use. EAM and RPH provided the cloud-sliced $NO_2$ dataset used for model evaluation. DBM and KCW provided the CrIS datasets of isoprene columns used for model evaluation. ABG provided feedback on the final results and the manuscript. All co-authors read and commented the first draft of the manuscript and provided feedback.

**Competing interests**

The authors declare that they have no conflict of interest.

**Data availability**


The CAMS-GLOB-ANTv5.3 (Granier et al., 2019) and CAMS-BIO-SOILv2.4 inventories (Simpson and Darras, 2021) are available on the ECCAD website at https://eccad.sedoo.fr (last access: 30 June 2024). Monthly LAI distributions from MODIS15A2H collection 6 are available at https://lpdaac.usgs.gov (last access: 30 June 2024). The Copernicus operational product TROPOMI S5p HCHO columns were downloaded from https://s5phub.copernicus.eu/dhus/#/home (last access: 30

June 2024). Monthly trace gases concentrations of $NO_2$ from the INDAAF network were downloaded from https://indaaf.obs-mip.fr/ (last access: 30 June 2024). Seasonal mean mixing ratios of the upper-troposphere $NO_2$ are available at https://maraisresearchgroup.co.uk/datasets.html (last access: 30 June 2024). Climatological means 2012-2020 of CrIS isoprene columns are available at https://conservancy.umn.edu/items/4fe84ff6-352c-4d8e-b777-b203aca870ba and 2019 data were provided by D. Millet. Monthly TROPOMI-based top-down inventories for 2019 soil and lightning $NO_x$ emissions can be

accessed from the BIRA-IASB Emission portal: https://emissions.aeronomie.be/index.php/tropomi-based/tropomi-soil-





## Acknowledgements


This research was performed as part of the EQUATOR project funded by the Belgian Science Policy Office (BELSPO) through the BRAIN-be 2.0 programme (contract no. B2/202/P1/EQUATOR, 2021-2025). It was also supported by the IMPALA project (contract no. 4000139771/22/I-DT-bgh, 2023-2025) funded by the European Space Agency, the PRODEX TROVA-E2 (2021-2023) and TROVA-3 (2024-2026) projects by the European Space Agency via BELSPO. DBM and KCW acknowledge

support from NASA (grant nos. 80NSSC20K0927 and 80NSSC24M0037).

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
