# Peer review of "Natural emissions of VOC and NOx over Africa constrained by TROPOMI HCHO and NO2 data using the MAGRITTEv1.1 model"

_EGUsphere, 2024_

## Author Comment (AC2)

**Reply to Referee#1**

We would like to thank the reviewer for their positive evaluation of the manuscript and for the useful comments and suggestions. Below we address the raised concerns. The reviewer's comments are shown in **bold** and our replies are given in blue. Additions to the original text are in green.

First, we applied the changes suggested in the Interactive Discussion CC1 'Citations for CAMS-GLOB-SOIL' by David Simpson. More specifically:

L.149-151: New references were added to the CAMS-BIO-SOILv2.4 inventory: (Simpson et al., 2023; Simpson and Segers, 2024). Also, we added Steinkamp and Lawrence (2011) and Yienger and Levy (1995) that replace Hudman et al. (2012).

Table 2: Reference to Darras et Simpson (2021) was replaced by Simpson et al. (2023).

Second, we add the following:

L.105: "A few other studies advocated for joint inversion and applied a similar approach (Souri et al., 2020, 2024; Wells et al., 2020)."

**The paper provides a comprehensive overview of the NOx and VOCs sources in sub-Saharan Africa. It is well written and has a clear outline.**

**There is a large increase in isoprene for region 1 and region 3 in the Northern Hemisphere during the dry season, peaking in March for SVOC, STD and ALBE inversions. At this time and location, the HCHO columns are quite high (figure 10), and this coincides with the peak in fire emissions, which must explain part of the increase in isoprene. At the same, these inversions do not capture the increase in isoprene during fall in region 1. While the precise source attribution will not be solved in this study, you could still provide some context at whether why the isoprene columns are not well represented in both the prior simulations and inversions and provide some hypothesi(e)s to guide future studies. A figure of the seasonal cycle of NOx and VOCs surface emissions by source (anthropogenic, biogenic and pyrogenic) for both the prior and the STD inversions (having the other inversions might make the plot too busy) could illustrate this issue.**

We added the Supplementary Section 6 containing Figure S6 that is referenced in the newly added paragraph of the manuscript. We add the following in the manuscript:

L.679: "The isoprene peak is not captured by any of the sensitivity inversions. It is worth noting that the interannual variability of CrIS columns in regions 1 and 2 is relatively weak (Fig. S6), except for year 2019, which exhibited enhanced isoprene levels in October, in particular over East Africa (Fig. S6(b)). The months from October to December 2019 were among the wettest on record in Sahel, West, Central and East Africa (Wainwright et al., 2021; WMO, 2019). These extreme conditions might have led to the low TROPOMI HCHO levels because of the reduced

photochemical activity and therefore to lower OH production, and to higher CrIS levels due to relatively longer isoprene lifetimes under cloudy conditions."

**Even though the increments on anthropogenic emissions are not large, it would be appreciated to add a key point about main takeaways on this subject in the conclusion section.**

We added the following sentence to the conclusion:

**L.734**: "Top-down anthropogenic emissions indicate localised reductions in $NO_X$ fluxes in Nigeria and in the Highveld region of South Africa, as well as decreased VOC fluxes (by about 20%) across Nigeria. Elsewhere, the changes are small or negligible. "

**Minor comments:**

**L.47: "Global bottom-up inventories for soil and lightning fluxes range from 1.3 to 6.6 TgN yr-1 (Murray, 2016; Vinken et al., 2014; Weng et al., 2020), and from 4 to 34 TgN yr-1 (Steinkamp and Lawrence, 2011; Yan et al., 2005), respectively". You switched up soil and lightning.**

Corrected.

**L.60: remove the parenthesis after "Lebel et al., 2011);"**

Done.

**L.63: "By virtue of their global coverage and continuous monitoring, spaceborne data are great alternatives to studying air composition in Africa and are often used to constrain model emission estimates obtained with chemistry-transport models (CTMs)." At this stage of the discussion, you could say: can be used to constrain instead of are often.**

Changed.

**L.98: first definition of TROPOMI and S5P acronyms, so no need to redefine them at line 177.**

Corrected.

**L.101: first instance of MAGRITTE, please define the acronym here.**

**L.101**: "For this purpose, the Model of Atmospheric composition at Global and Regional scales using Inversion Techniques for Trace Gas Emissions (MAGRITTEv1.1) chemistry-transport model (Müller et al., 2019)…"

**L.114**: We removed the definition of MAGRITTE since it is now defined in L.101.

**L.105: make it 3 sentences for section 2, 3 and 4. Please be consistent and choose either Sect. or Section.**

The changes were applied in **L.110-111**.

**2.1 MAGRITTEv1.1 chemistry-transport model: If you are using a limited area version of the model, you should specify how you prescribed the boundary conditions.**

We agree. This is now clarified as follows.

**L.117**: "The MAGRITTE model is first run in its global configuration at a spatial resolution of $2° \times 2.5°$ (latitude, longitude). This simulation provides the boundary conditions for the regional MAGRITTEv1.1 model that covers the African domain delimited by 30°S-17°N and 18°W-56°E with a $0.5° \times 0.5°$ horizontal resolution, between the surface and the lower stratosphere with 40 hybrid sigma-pressure levels in the vertical."

**Reference**

Simpson, D. and Segers, A.: Transboundary particulate matter, photo-oxidants, acidifying and eutrophying components. EMEP Status Report 1/2024, The Norwegian Meteorological Institute, Oslo, Norway, 2024.

Simpson, D., Benedictow, A., and Darras, S.: The CAMS soil emissions: CAMS-GLOB-SOIL, in: CAMS2_61 – Global and European emission inventories., in: Documentation of CAMS emission inventory products, vol. Chap. 9, 59–70, 2023.

Souri, A. H., Nowlan, C. R., González Abad, G., Zhu, L., Blake, D. R., Fried, A., Weinheimer, A. J., Wisthaler, A., Woo, J.-H., Zhang, Q., Chan Miller, C. E., Liu, X., and Chance, K.: An inversion of $NO_x$ and non-methane volatile organic compound (NMVOC) emissions using satellite observations during the KORUS-AQ campaign and implications for surface ozone over East Asia, Atmospheric Chem. Phys., 20, 9837–9854, https://doi.org/10.5194/acp-20-9837-2020, 2020.

Souri, A. H., González Abad, G., Wolfe, G. M., Verhoelst, T., Vigouroux, C., Pinardi, G., Compernolle, S., Langerock, B., Duncan, B. N., and Johnson, M. S.: Feasibility of robust estimates of ozone production rates using satellite observations, EGUsphere, 1–36, https://doi.org/10.5194/egusphere-2024-1947, 2024.

Steinkamp, J. and Lawrence, M. G.: Improvement and evaluation of simulated global biogenic soil NO emissions in an AC-GCM, Atmospheric Chem. Phys., 11, 6063–6082, https://doi.org/10.5194/acp-11-6063-2011, 2011.

Wainwright, C. M., Finney, D. L., Kilavi, M., Black, E., and Marsham, J. H.: Extreme rainfall in East Africa, October 2019–January 2020 and context under future climate change, Weather, 76, 26–31, https://doi.org/10.1002/wea.3824, 2021.

Wells, K. C., Millet, D. B., Payne, V. H., Deventer, M. J., Bates, K. H., de Gouw, J. A., Graus, M., Warneke, C., Wisthaler, A., and Fuentes, J. D.: Satellite isoprene retrievals constrain emissions and atmospheric oxidation, Nature, 585, 225–233, https://doi.org/10.1038/s41586-020-2664-3, 2020.

WMO: State of the Climate in Africa, World Meteorological Organization (WMO), 2019.

Yienger, J. J. and Levy, H.: Empirical model of global soil-biogenic $NO_\chi$ emissions, J. Geophys. Res. Atmospheres, 100, 11447–11464, https://doi.org/10.1029/95JD00370, 1995.

---

## Author Comment (AC3)

**Reply to Referee#2**

We would like to thank the reviewer for their positive evaluation of the manuscript and for the useful comments and suggestions. Below we address the raised concerns. The reviewer's comments are shown in **bold** and our replies are given in blue. Additions to the original text are in green.

First, we applied the changes suggested in the Interactive Discussion CC1 'Citations for CAMS-GLOB-SOIL' by David Simpson. More specifically:

**L.149-151**: New references were added to the CAMS-BIO-SOILv2.4 inventory: (Simpson et al., 2023; Simpson and Segers, 2024). Also, we added Steinkamp and Lawrence (2011) and Yienger and Levy (1995) that replace Hudman et al. (2012).

Table 2: Reference to Darras et Simpson (2021) was replaced by Simpson et al. (2023).

Second, we add the following:

**L.105**: "A few other studies advocated for joint inversion and applied a similar approach (Souri et al., 2020, 2024; Wells et al., 2020)."

**The authors conducted adjoint-based inverse modelling to update emissions of NOx and VOC over Africa for 2019 using a chemistry-transport model MAGRITTEv1.1 and TROPOMI HCHO and NO₂ retrievals. This is a robust study. Even though there are uncertainties associated with assumptions and methods in the approach, the authors conduct additional analyses (an ensemble of inversions) to evaluate and address these uncertainties. The updated emissions have also been evaluated using available observations. This study is important and informative given limited knowledge and observations of Africa NOx and VOC emissions. The manuscript is well written. I have no major concerns or comments on this manuscript. Below are only a few minor suggestions.**

**Please consider adding latitude and longitude to your map plots. For example, Line 393 says "decreases in soil emissions in the Northern Hemisphere". Without latitude and longitude, it's hard to visualise in Figure 11.**

For the sake of clarity, we have emphasised the coordinate lines in Fig. 6. As figures already contain a substantial amount of information, we believe that including latitudes and longitudes would further reduce their readability.

**Caption of Figure 17: Please change "Histogram of average soil NO flux measurements" to "Average soil NO flux".**

Done.

**I wonder if the author could add more discussion on the implication on MEGAN.**

We add the Supplementary Section 7 containing the Table S2 referenced in the paragraph we add to the manuscript. We add the following to the manuscript:

**L.689:** "Our results point to a likely underestimation of basal isoprene emissions of the species in southern Africa and tropical western Africa, as also underscored by Marais et al. (2014). The STD inversion uses the MEGAN bottom-up inventory (Sect. 2.1.1 and Fig. 3), which relies on emission capacity maps derived from detailed ecoregion descriptions. These maps integrate information on species composition with species-specific emission factors. However, these maps bear uncertainties related to the species composition estimates and to the basal emission capacities that stem from several factors, including within-species, diurnal and seasonal variabilities, measurement and analysis errors, and differences between sun- and shade-adapted leaves. Despite these difficulties, a single value of the basal emission capacity is assigned for the entire canopy and the entire year, typically based on measurements in the summer for shade-adapted leaves. Based on the inversion results, the derived average isoprene emission capacity in regions 1 and 2 is relatively close to the a priori (Table S2), whereas the emission capacity for broadleaf forests is increased by about 20% (from 4,100 to 5,100 $\mu g\ m^{-2}\ h^{-1}$) and the emission capacity of savannas in region 4 is strongly enhanced after optimisation, from 2,500 $\mu g\ m^{-2}\ h^{-1}$ to 6,000 $\mu g\ m^{-2}\ h^{-1}$. High emission rates in southern African woodlands, notably from *Diplorhynchus condylocarpon* and the Miombo Woodlands, were reported in Otter et al. (2003), in a region encompassing southern Angola and northern Mozambique (eastern Miombo woodlands), in line with our results."

Also, we add a paragraph in the conclusion section about future research direction:

**L. 785:** "Building on the work of Marais et al. (2014), future research could focus on deriving a map of emission potential developed for each ecosystem based on top-down isoprene emission estimates. With recent advances in satellite-based isoprene estimates, top-down inventories could be employed to assign MEGAN isoprene emission capacities in landscapes where data on species composition and species-specific emission factors are sparse."

**Reference**

Marais, E. A., Jacob, D. J., Guenther, A., Chance, K., Kurosu, T. P., Murphy, J. G., Reeves, C. E., and Pye, H. O. T.: Improved model of isoprene emissions in Africa using Ozone Monitoring Instrument (OMI) satellite observations of formaldehyde: implications for oxidants and particulate matter, Atmospheric Chem. Phys., 14, 7693–7703, https://doi.org/10.5194/acp-14-7693-2014, 2014.

Otter, L., Guenther, A., Wiedinmyer, C., Fleming, G., Harley, P., and Greenberg, J.: Spatial and temporal variations in biogenic volatile organic compound emissions for Africa south of the equator, J. Geophys. Res. Atmospheres, 108, https://doi.org/10.1029/2002JD002609, 2003.

Simpson, D. and Segers, A.: Transboundary particulate matter, photo-oxidants, acidifying and eutrophying components. EMEP Status Report 1/2024, The Norwegian Meteorological Institute, Oslo, Norway, 2024.

Simpson, D., Benedictow, A., and Darras, S.: The CAMS soil emissions: CAMS-GLOB-SOIL, in: CAMS2_61 – Global and European emission inventories., in: Documentation of CAMS emission inventory products, vol. Chap. 9, 59–70, 2023.

Souri, A. H., Nowlan, C. R., González Abad, G., Zhu, L., Blake, D. R., Fried, A., Weinheimer, A. J., Wisthaler, A., Woo, J.-H., Zhang, Q., Chan Miller, C. E., Liu, X., and Chance, K.: An inversion of $NO_x$ and non-methane volatile organic compound (NMVOC) emissions using satellite observations during the KORUS-AQ campaign and implications for surface ozone over East Asia, Atmospheric Chem. Phys., 20, 9837–9854, https://doi.org/10.5194/acp-20-9837-2020, 2020.

Souri, A. H., González Abad, G., Wolfe, G. M., Verhoelst, T., Vigouroux, C., Pinardi, G., Compernolle, S., Langerock, B., Duncan, B. N., and Johnson, M. S.: Feasibility of robust estimates of ozone production rates using satellite observations, EGUsphere, 1–36, https://doi.org/10.5194/egusphere-2024-1947, 2024.

Steinkamp, J. and Lawrence, M. G.: Improvement and evaluation of simulated global biogenic soil NO emissions in an AC-GCM, Atmospheric Chem. Phys., 11, 6063–6082, https://doi.org/10.5194/acp-11-6063-2011, 2011.

Wells, K. C., Millet, D. B., Payne, V. H., Deventer, M. J., Bates, K. H., de Gouw, J. A., Graus, M., Warneke, C., Wisthaler, A., and Fuentes, J. D.: Satellite isoprene retrievals constrain emissions and atmospheric oxidation, Nature, 585, 225–233, https://doi.org/10.1038/s41586-020-2664-3, 2020.

Yienger, J. J. and Levy, H.: Empirical model of global soil-biogenic $NO_\chi$ emissions, J. Geophys. Res. Atmospheres, 100, 11447–11464, https://doi.org/10.1029/95JD00370, 1995.